# Clinical value of the fibrosis-4 index in predicting mortality in patients with right ventricular pacing

Naoya Inoue[1,2]*, Shuji Morikawa[1,2], Takashi Ogane[1], Takehiro Hiramatsu[1,2], Toyoaki Murohara[2]

1 Department of Cardiology, Chutoen General Medical Center, Kakegawa, Shizuoka, Japan, 2 Department of Cardiology, Nagoya University Graduate School of Medicine, Nagoya, Aichi, Japan

* inouenaoya251410@gmail.com

## Abstract

### Background

The fibrosis-4 (FIB-4) index has attracted attention as a predictive factor for cardiovascular events and mortality in patients with heart disease. However, its clinical value in patients with implanted pacemakers remains unclear.

### Methods

This study included patients who underwent pacemaker implantation. The FIB-4 index was calculated based on blood tests performed during the procedure. The primary outcome was all-cause mortality, and the secondary outcomes included cardiovascular death, non-cardiovascular death, and major adverse cardiovascular events (MACE; composite of cardiovascular death, heart failure hospitalization, non-fatal myocardial infarction, and non-fatal stroke). The FIB-4 index was stratified into tertiles. Between-group comparisons were performed using log-rank tests and multivariate analysis using Cox proportional hazards. The predictive accuracy and cut-off value of the FIB-4 index were calculated from the receiver operating characteristic curve for all-cause mortality. Finally, based on the calculated cut-off values, the patients were divided into two groups for outcome validation and subgroup analysis.

### Results

This study included 201 participants, of whom 38 experienced death during the observation period (median: 1097 days). All-cause mortality, non-cardiovascular death, and MACE differed significantly between groups stratified by the FIB-4 index tertiles (log-rank test: P<0.001, P<0.001, and P = 0.045, respectively). Using Cox proportional hazards analysis, the unadjusted hazard ratio was 4.75 (95% confidence interval [CI]: 2.05–11.0, P<0.001) for Tertile 3 compared to Tertile 1. After adjustment for confounding factors, including sex, the presence or absence of left bundle branch block at baseline, QRS duration during pacing, and pacing rate

---

**Funding:** The author(s) received no specific funding for this work.

**Competing interests:** The authors have declared that no competing interests exist.

at the last check, the hazard ratio was 4.79 (95% CI: 2.04–11.2, *P*<0.001). The cut-off value of the FIB-4 index was 3.75 (area under the curve: 0.72, 95% CI: 0.62–0.82).

## Conclusions

In patients with pacemakers, the FIB-4 index may be a predictor of early all-cause mortality, with a cut-off value of 3.75.

## Introduction

In the past, right ventricular pacing (RVP) was used for bradycardia arrhythmias; however, it was associated with problems such as decreased left ventricular systolic function [1], increased hospitalization for heart failure [2], and the onset of atrial fibrillation [3] due to electrical and mechanical dyssynchrony. To address these issues, pacing techniques involving the His bundle or the left bundle area have recently been introduced [4, 5]. Many studies have compared the outcomes of these novel pacing techniques with those of conventional RVP [6] and investigated the factors contributing to the problems associated with RVP, with pacing rate and QRS duration reported as problematic factors [7, 8]. However, to the best of our knowledge, studies of the relationship between RVP and mortality are limited. The fibrosis-4 (FIB-4) index, a non-invasive marker used to assess the degree of liver fibrosis, has been reported as a predictive factor for right ventricular function and major adverse cardiovascular events in patients with heart diseases, such as heart failure and tricuspid regurgitation [9, 10]. In a study targeting heart failure in patients with preserved left ventricular systolic function, patients with an FIB-4 index ≥ 3.11 had a 2.202-fold (95% confidence interval: 1.110–4.368) higher risk of major adverse cardiovascular events (composite of cardiovascular death, heart failure-related rehospitalization, non-fatal myocardial infarction, and non-fatal stroke) compared to patients with an index <3.11 after adjustment for male sex, serum creatinine level, and hemoglobin concentration [10]. The FIB-4 index is related to cardiovascular events because prolonged hepatic congestion due to reduced blood flow and sodium diuresis leads to fluid retention and arterial sclerosis, which impair the left ventricular diastolic function [10, 11]. However, to date, no study has investigated the prognosis of patients with right ventricular pacemakers in relation to the FIB-4 index. Therefore, this study aimed to investigate whether the FIB-4 index is associated with mortality in patients with right ventricular pacemakers. Patients requiring pacemaker implantation often have circulatory insufficiency and congestion at the time of bradycardia arrhythmia onset, and many have elevated B-type natriuretic peptide (BNP) levels. If the FIB-4 index can be confirmed to predict prognosis, regardless of the pacing factors, it may have high clinical value.

## Materials and methods

### Patients

This retrospective, single-center, observational study included patients who underwent right ventricular pacemaker implantation for sinus node dysfunction or atrioventricular block between April 2015 and December 2021. The following patients were excluded: 1) those who underwent stimulation conduction system pacemaker implantation, such as His bundle pacing or left bundle area pacing; 2) those with a ventricular pacing rate of < 1% at the final pacemaker check; 3) those who did not undergo follow-up at our hospital after right ventricular

pacemaker implantation; 4) those for whom right ventricular electrocardiograms were unobtainable (including only fusion waveforms); and 5) those who underwent VVI-pacemaker implantation for bradycardia-induced atrial fibrillation.

The study was conducted in accordance with the standards of the Declaration of Helsinki and current ethical guidelines and was approved by the Ethics Committee on Medical Research of Chutoen General Medical Center (reference no: 1241230822). As this study was not a clinical trial and the data were retrospectively and anonymously collected and analyzed, the requirement for patients' written informed consent was waived.

The data were accessed for research purposes on August 23, 2023. The authors had access to information that could identify individual participants during and after the data collection.

## Data collection

The FIB-4 index was calculated based on the patient's age and blood test results at the time of pacemaker implantation using the following formula: FIB-4 index = age × aspartate aminotransferase [AST] level / (platelets × square root [alanine transaminase level]). The study participants were divided into three groups based on their FIB-4 index values: Tertile 1 (FIB-4 ≤ 2.17; n = 67), Tertile 2 (2.18 ≤ FIB-4 ≤ 3.28; n = 67), and Tertile 3 (FIB-4 ≥ 3.29; n = 67). Patient characteristics were compared between the groups. In addition, the albumin–bilirubin (ALBI) score was calculated using the following formula: ALBI score = ((log 10 (bilirubin [mg/dL] × 17.1) × 0.66) + (albumin [g/dL] ×10 × [−0.0852]).

## Primary and secondary outcomes

The primary outcome of this study was all-cause mortality, which included deaths due to cardiovascular disease, cancer, infections, and other causes. The secondary outcomes were cardiovascular death, non-cardiovascular death, and major adverse cardiovascular events (MACE). MACE included cardiovascular death, which comprised deaths resulting from arrhythmia, myocardial infarction, heart failure, or stroke, and heart failure hospitalization, non-fatal myocardial infarction, or non-fatal stroke. Non-cardiovascular death was defined as deaths caused by non-cardiovascular factors, such as cancer, infection, and renal failure.

## Electrocardiogram analysis

The 12-lead electrocardiograms (ECGs) obtained preoperatively and during pacemaker checks were analyzed. The ECGs were recorded at a paper speed of 25 mm/s and an amplitude of 10 mm/mV. Two cardiologists independently evaluated the following parameters from each ECG: 1) heart rate; 2) QRS duration (during bradycardia, arrhythmias, and pacing); and 3) the presence or absence of conduction abnormalities.

## Right ventricular pacemaker implantation and pacemaker check

Every patient underwent implantation of a dual-chamber pacemaker in the right ventricle (RV). The specific site for RV lead implantation was the RV septum, which was aimed at under fluoroscopic guidance. Nonetheless, the ultimate determination regarding lead placement was at the discretion of the principal doctor, who considered the optimal electrical parameters and lead stability.

Pacemaker assessments were conducted prior, and subsequent, to implantation. After discharge, patients received regular outpatient follow-ups at intervals of 6 to 9 months. During these visits, appropriate ECG and pacemaker parameters were documented.

## Statistical analysis

Regarding the statistical analysis, categorical variables were analyzed using the chi-square or Fisher's exact tests and presented as numbers and percentages. Continuous variables were analyzed using non-parametric tests (Kruskal–Wallis test), and if necessary, the Mann–Whitney U test was performed. The occurrence of mortality events during the post-implantation follow-up period was computed, and the curves depicting the incidence of these events were compared between groups using log-rank assessment. Owing to suspected deviations from proportional hazards in certain analyses, additional analyses were conducted with an observation period of up to 1000 days. In the comparison of patient characteristics among the three groups, we conducted the Kruskal-Wallis test and reported the P-value for overall comparison. To adjust for the significance level in multiple comparisons, we performed post-hoc analyses using the Bonferroni correction.

Univariate analysis using Cox proportional hazards was conducted for both the primary and secondary outcomes. Multivariate analysis, adjusted for sex and ALBI score, was performed to determine the primary outcomes. Furthermore, multivariate analysis using the forced entry method was performed for other potential factors influencing clinical outcomes, such as pacing rate during RVP, QRS duration, estimated glomerular filtration rate (eGFR), BNP level, and tricuspid regurgitation pressure gradient (TR-PG). Furthermore, after identifying the cutoff value for all-cause mortality from the receiver operating characteristic curve, we categorized the groups based on this cut-off and ultimately examined whether it excelled in predicting events in conjunction with other predictive factors in multivariate analysis.

Statistical significance was set at $P < 0.05$. All statistical analyses were performed using R with EZR software version 1.62 (Saitama Medical Center, Jichi Medical University, Saitama, Japan) [12]. Regarding missing data, only the samples with complete data for individual statistical analyses were included.

## Results

Among the patients who underwent pacemaker implantation at our hospital between April 2015 and December 2021, 201 were included in the final analysis. In the comparison of the FIB-4 index tertiles, the median FIB-4 index values were as follows: Tertile 1, 1.80; Tertile 2, 2.66; and Tertile 3, 4.19 ($P < 0.001$). In addition to the elements of the FIB-4 index, such as age, AST, and platelet count, significant differences were observed in eGFR, hemoglobin, low-density lipoprotein cholesterol, BNP levels, and dyslipidemia (Table 1). The number of patients with a positive HCV Ab was significantly different among the three groups; however, post hoc analysis using the Bonferroni method revealed no significant differences between each pair of groups (Tertile 1 vs 2; $P = 1$, Tertile 1 vs 3; $P = 0.091$, Tertile 2 vs 3; $P = 0.23$). Regarding cancer, a cancer related to the liver was not observed as one of the CV risk factors. While a significant overall difference was noted for cancer, Bonferroni analysis showed no significant differences between Tertile 1 vs 2; $P = 1$, Tertile 1 vs 3; $P = 0.62$, Tertile 2 vs 3; $P = 0.076$.

No significant differences were observed in terms of treatment at baseline or pacemaker parameters at the last follow-up. However, there was a significant difference in the TR-PG on the cardiac ultrasound examination and the proportion of left bundle branch block (LBBB) on the baseline ECG (Table 1).

Among the study patients (median age: 81 years), 38 died during the observation period (median: 1097 days), resulting in an incidence rate of 5.8/100 person-years. The incidence rates for all-cause mortality, stratified into three groups based on the FIB-4 index, were as

**Table 1. Patient characteristics according to the FIB-4 index tertiles.**

| | Total (n = 201) | Tertile 1 FIB-4 ≤ 2.17 (n = 67) | Tertile 2 2.18≤FIB-4≤3.28 (n = 67) | Tertile 3 FIB-4 ≥ 3.29 (n = 67) | *P*-value |
|---|---|---|---|---|---|
| Age, years | 81 (73–87) | 71 (67–80) | 82 (76–87) | 86 (80–89) | <0.001 |
| Sex, male, n (%) | 102 (50) | 33 (49) | 35 (52) | 34 (50) | 0.98 |
| **Laboratory data** | | | | | |
| Albumin, g/dL | 3.8 (3.4–4.0) | 3.9 (3.7–4.3) | 3.8 (3.5–4.0) | 3.5 (3.3–3.8) | <0.001 |
| T-bilirubin, mg/dL | 0.7 (0.6–0.9) | 0.7 (0.6–0.9) | 0.8 (0.6–0.9) | 0.7 (0.6–0.9) | 0.49 |
| ALBI score | -2.48 (-2.74- -2.20) | -2.61 (-2.91- -2.45) | -2.49 (-2.70- -2.23) | -2.27 (-2.48- -2.08) | <0.001 |
| AST, IU/L | 26 (20–35) | 22 (18–27) | 26 (19–31) | 33 (25–60) | <0.001 |
| ALT, IU/L | 20 (13–31) | 21 (14–29) | 17 (13–27) | 26 (13–53) | 0.083 |
| eGFR, mL/min/1.73 m$^2$ | 53.3 (39.1–63.8) | 61.1 (45.7–72.0) | 50.7 (40.8–63.7) | 46.4 (34.2–56.8) | 0.001 |
| Hb, g/L | 12.5 (11.1–13.8) | 13.3 (11.9–14.5) | 12.7 (11.3–14.0) | 11.6 (10.3–12.9) | <0.001 |
| Platelet count, 10$^9$/L | 18.6 (15.1–21.5) | 21.4 (19.0–24.5) | 18.0 (15.6–21.0) | 14.8 (12.0–18.1) | <0.001 |
| FIB-4 index | 2.66 (1.98–3.61) | 1.80 (1.30–1.98) | 2.66 (2.39–2.97) | 4.19 (3.62–4.88) | <0.001 |
| LDL-C, mg/dL | 100 (87–118) | 108 (93–130) | 95 (82–114) | 96 (82–114) | 0.001 |
| HbA1c, % | 6.1 (5.8–6.5) | 6.1 (5.9–6.6) | 6.0 (5.7–6.3) | 6.0 (5.7–6.7) | 0.080 |
| BNP, pg/L | 169 (58–464) | 97 (46–213) | 224(60–498) | 369(118–592) | <0.001 |
| HBs Ag status Positive, n (%) | 4 (1) | 2 (2) | 2 (2) | 0 (0) | 0.54 |
| HCV Ab status Positive, n (%) | 15 (7) | 2 (2) | 3 (4) | 10 (14) | 0.023 |
| **CV risk factors** | | | | | |
| Hypertension, n (%) | 130 (64) | 44 (65) | 46 (68) | 40 (59) | 0.58 |
| Diabetes mellitus, n (%) | 61 (30) | 22 (32) | 21 (31) | 18 (26) | 0.79 |
| Dyslipidemia, n (%) | 45 (22) | 22 (32) | 13 (19) | 10 (14) | 0.04 |
| Smoking, n (%) | 78 (38) | 25 (37) | 32 (47) | 21 (31) | 0.15 |
| Post-PCI, n (%) | 17 (8) | 6 (8) | 3 (4) | 8 (11) | 0.33 |
| Prior MI, n (%) | 12 (5) | 3 (4) | 3 (4) | 6 (8) | 0.60 |
| AF, n (%) | 39 (19) | 15 (22) | 14 (20) | 10 (14) | 0.52 |
| Stroke, n (%) | 27 (13) | 12 (17) | 10 (14) | 5 (7) | 0.18 |
| COPD, n (%) | 16 (7) | 5 (7) | 5 (7) | 6 (8) | 1 |
| Cancer, n (%) | 36 (17) | 11 (16) | 7 (10) | 18 (26) | 0.047 |
| **Treatment at baseline** | | | | | |
| ACEI/ARB, n (%) | 103 (51) | 33 (49) | 37 (55) | 33 (49) | 0.74 |
| Beta-blockers, n (%) | 21 (10) | 6 (8) | 8 (11) | 7 (10) | 0.95 |
| CCB, n (%) | 100 (49) | 30 (44) | 35 (52) | 35 (52) | 0.64 |
| MRA, n (%) | 14 (6) | 2 (2) | 8 (11) | 4 (5) | 0.14 |
| Diuretics, n (%) | 50 (24) | 14 (20) | 17 (25) | 19 (28) | 0.63 |
| Statins, n (%) | 58 (28) | 26 (38) | 18 (26) | 14 (20) | 0.07 |
| Antiplatelets, n (%) | 39 (19) | 18 (26) | 11 (16) | 10 (14) | 0.20 |
| Oral anticoagulants, n (%) | 29 (14) | 9 (13) | 14 (20) | 6 (8) | 0.16 |
| **Echocardiogram data at baseline** | | | | | |
| LAD size, mm | 41 (37–45) | 41 (36–44) | 42 (38–45) | 41 (37–47) | 0.33 |
| LVEF, % | 67 (60–72) | 67 (59–72) | 67 (59–71) | 67 (63–72) | 0.79 |
| LVDd, mm | 47 (43–51) | 47 (44–51) | 46 (43–50) | 47 (43–52) | 0.45 |
| LVDs, mm | 29 (26–32) | 29 (27–32) | 29 (26–32) | 29 (26–33) | 0.88 |
| TR-PG, mmHg | 26.8 (24.0–30.4) | 26.1 (23.4–29.9) | 26.8 (24.5–29.9) | 29.0 (25.2–35.5) | 0.007 |
| **ECG at baseline** | | | | | |
| SND, n (%) | 31 (15) | 12 (17) | 16 (23) | 3 (4) | 0.004 |
| AVB, n (%) | 170 (84) | 55 (82) | 51 (76) | 64 (95) | |

*(Continued)*

**Table 1.** (Continued)

| | Total (n = 201) | Tertile 1 FIB-4 ≤ 2.17 (n = 67) | Tertile 2 2.18≤FIB-4≤3.28 (n = 67) | Tertile 3 FIB-4 ≥ 3.29 (n = 67) | *P*-value |
|---|---|---|---|---|---|
| QRS duration, ms | 120 (90–140) | 122 (90–140) | 118 (91–136) | 122 (90–141) | 0.78 |
| RBBB, n (%) | 94 (46) | 33 (49) | 33 (49) | 28 (41) | 0.64 |
| LBBB, n (%) | 16 (7) | 5 (7) | 1 (1) | 10 (14) | 0.016 |
| LAHB, n (%) | 35 (17) | 11 (16) | 12 (17) | 12 (17) | 1 |
| LPHB, n (%) | 10 (4) | 5 (7) | 3 (4) | 2 (2) | 0.61 |
| ECG after RVP implantation | | | | | |
| HR, bpm | 69 (63–78) | 69 (63–76) | 69 (62–76) | 72 (63–81) | 0.36 |
| QRS duration, ms | 148 (138–160) | 148 (136–160) | 148 (138–158) | 150 (138–163) | 0.45 |
| RVP parameters at final check | | | | | |
| Atrial | | | | | |
| Pacing rate, % | 20 (5–48) | 17 (6–40) | 29 (6–55) | 15 (4–45) | 0.17 |
| P-wave amplitude, mV | 2.6 (1.6–3.9) | 2.6 (1.7–4.0) | 2.5 (1.4–4.0) | 2.7 (1.6–3.7) | 0.96 |
| Threshold, V | 0.6 (0.5–0.8) | 0.7 (0.5–0.9) | 0.6 (0.5–0.8) | 0.6 (0.5–0.7) | 0.15 |
| Impedance, Ω | 456 (399–539) | 470 (418–540) | 448 (399–520) | 437 (380–560) | 0.20 |
| Ventricular | | | | | |
| Pacing rate, % | 99 (67–100) | 98 (51–100) | 99 (27–100) | 99 (95–100) | 0.06 |
| R-wave amplitude, mV | 10.6 (7.9–14.8) | 11.0 (7.9–14.8) | 9.8 (7.6–14.5) | 11.8 (7.8–14.8) | 0.79 |
| Threshold, V | 0.8 (0.7–1.0) | 0.8 (0.6–1.0) | 0.8 (0.7–1.0) | 0.8 (0.6–1.0) | 0.51 |
| Impedance, Ω | 494 (440–571) | 494 (437–565) | 499 (458–585) | 494 (433–573) | 0.83 |

Data are presented as medians (interquartile ranges) for continuous variables and number of patients (n) and percentage (%). ACEI, angiotensin-converting enzyme inhibitor; AF, atrial fibrillation; ALBI, albumin–bilirubin; ALT, alanine aminotransferase; ARB, angiotensin receptor blocker; AST, aspartate aminotransferase; AVB, atrioventricular block; BNP, brain natriuretic protein; CCB, calcium channel blocker; COPD, chronic obstructive pulmonary disease; CV, cardiovascular; eGFR, estimated glomerular filtration rate; FIB-4, fibrosis-4; Hb, hemoglobin; HBs Ag, hepatitis B s antigen; HCV Ab, hepatitis C virus antibody; HR, heart rate; LAD, left atrium diameter; LAHB, left anterior hemiblock; LBBB, left bundle branch block; LDL, low-density lipoprotein; LPHB, left posterior hemiblock; LVDd, left ventricular end-diastolic diameter; LVDs, left ventricular end-systolic diameter; LVEF, left ventricular ejection fraction; MRA, mineralocorticoid receptor antagonist; OMI, old myocardial infarction; PCI, percutaneous coronary intervention; RBBB, right bundle branch block; SND, sinus node dysfunction; T-bilirubin, total-bilirubin; TR-PG, tricuspid regurgitant pressure gradient.

follows: Tertile 1, 2.9/100 person-years; Tertile 2, 2.3/100 person-years; and Tertile 3, 13.3/100 person-years (log-rank test, *P* < 0.001; Fig 1A and Table 2).

Between-group comparisons for the secondary outcomes, including cardiovascular, non-cardiovascular death, and MACE were then performed. The incidence of cardiovascular death was as follows: Tertile 1, 4/67 (5%); Tertile 2, 2/67 (2%); and Tertile 3, 7/67 (10%; log-rank test, *P* = 0.17; Fig 1B and Table 2). In contrast, the incidence of non-cardiovascular death was as follows: Tertile 1, 3/67 (4%); Tertile 2, 3/67 (4%); Tertile 3, 19/67 (28%), which was significantly different between the groups (log-rank test, *P* < 0.001; Fig 1C and Table 2). There was also a statistically significant difference in MACE (log-rank test, *P* = 0.045; Fig 1D). Details of cardiovascular death, non-cardiovascular death, and other major adverse cardiovascular events are shown in Table 3.

In the Cox proportional hazards analysis for all-cause mortality, based on the FIB-4 index with Tertile 1 as the reference group, a significant increase in risk was observed in Tertile 3, with an unadjusted hazard ratio of 4.75 (95% CI: 2.05–11.0; *P* < 0.001; Table 4).

Subsequently, the adjusted hazard ratio, considering clinically important factors, such as sex, ALBI score, and the presence or absence of LBBB at baseline; along with pacemaker-

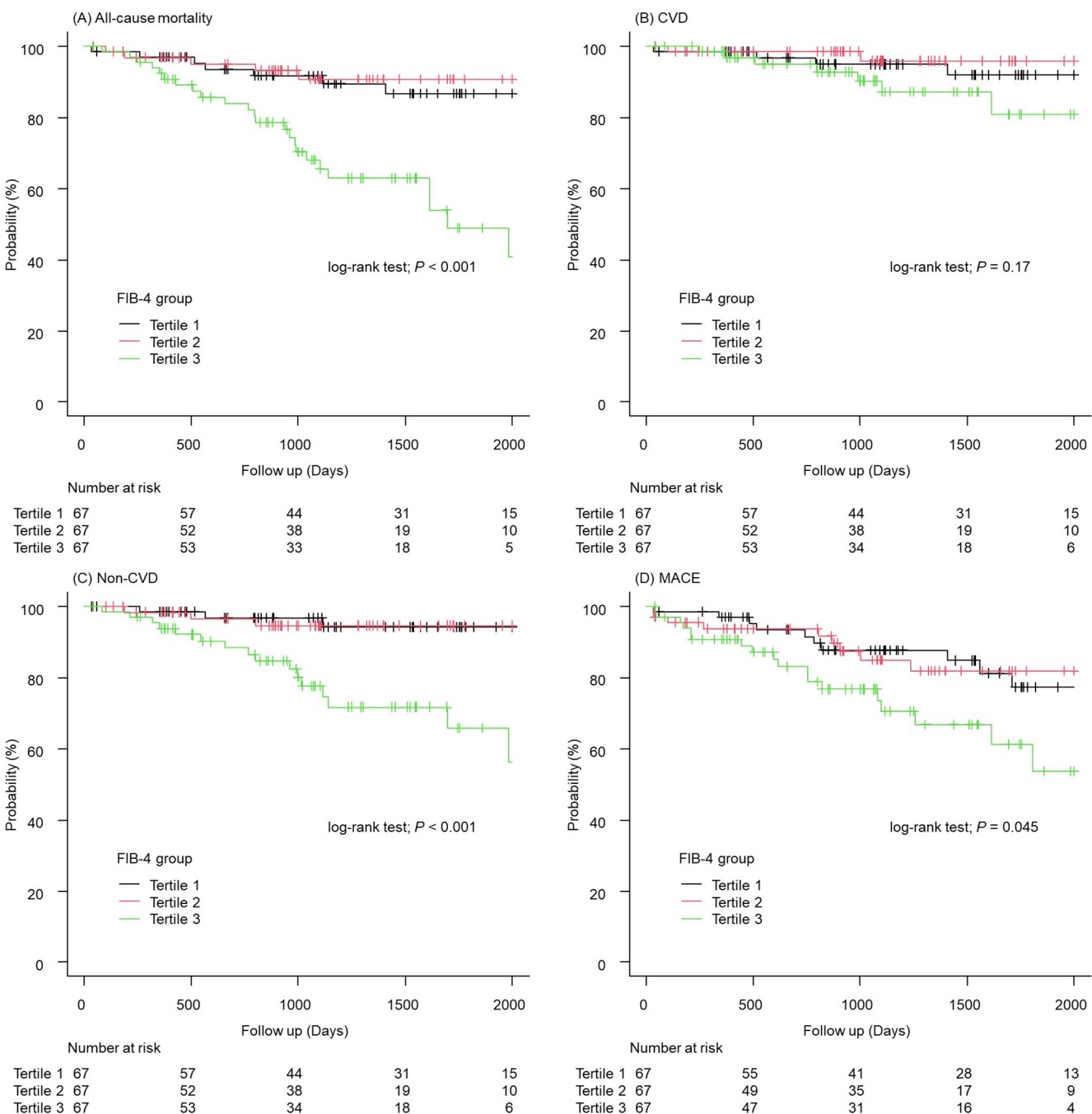

**Fig 1. All-cause mortality and secondary outcomes stratified by FIB-4 index tertiles.** (A) Kaplan–Meier curve of all-cause mortality. (B) Cardiovascular death. (C) Non-cardiovascular death. (D) MACE. CVD, Cardiovascular death; FIB-4, Fibrosis-4; MACE, Major adverse cardiovascular events.

related factors, such as QRS duration during pacing and ventricular pacing rate, was 3.67 (95% CI: 1.51–8.90; $P$ = 0.003) and 4.79 (95% CI: 2.04–11.2; $P$ < 0.001) for Tertile 3 and Tertile 1, respectively (Table 4). In addition, the adjusted hazard ratio considering factors related to congestion, such as BNP, eGFR, and TR-PG, was 3.48 (95% CI: 1.38–8.78; $P$ = 0.007; Table 4). However, the hazard ratio was 2.10 (95% CI: 0.79–5.55, $P$ = 0.13) when adjusted for ALBI score, BNP, and hemoglobin level, which were significantly different in the comparison of

**Table 2. Primary and secondary outcomes.**

| Outcome | Total (n = 201) | | Tertile 1 (n = 67) | | Tertile 2 (n = 67) | | Tertile 3 (n = 67) | |
|---|---|---|---|---|---|---|---|---|
| | Events | Incidence rate (/100 person-year) | Events | Incidence rate (/100 person-year) | Events | Incidence rate (/100 person-year) | Events | Incidence rate (/100 person-year) |
| **Primary outcome** | | | | | | | | |
| **All-cause mortality** | 38 | 5.8 | 7 | 2.9 | 5 | 2.3 | 26 | 13.3 |
| **Secondary outcome** | | | | | | | | |
| **CVD** | 13 | 2.0 | 4 | 1.6 | 2 | 0.9 | 7 | 3.5 |
| **Non-CVD** | 25 | 3.8 | 3 | 1.2 | 3 | 1.4 | 19 | 9.6 |
| **HF hospitalization** | 24 | 3.8 | 7 | 2.9 | 3 | 1.4 | 14 | 7.9 |
| **Non-fatal MI** | 2 | 0.3 | 1 | 0.4 | 1 | 0.4 | 0 | 0 |
| **Non-fatal stroke** | 6 | 0.9 | 1 | 0.4 | 3 | 1.4 | 2 | 1.0 |

Data are presented as the number of patients (n). CVD, Cardiovascular death; FIB-4, Fibrosis-4; HF, Heart failure; MI, myocardial infarction.

patient characteristics (Table 4). To assess the impact of adjustment factors, Cox analyses were performed using multiple combinations of models for the ALBI score, BNP level, and hemoglobin level (Table 5). The results indicated that each factor influenced prognosis, with a specific focus on examining the differences between BNP and FIB-4 index as predictive factors. To achieve this, BNP was stratified into tertiles, and Kaplan-Meier analysis with log-rank tests was conducted (Fig 2). The log-rank test results showed significance for both the FIB-4 index

**Table 3. Details of cardiovascular death, non-cardiovascular death, and other major adverse cardiovascular events.**

| Outcome | Events No. |
|---|---|
| **Primary outcome** | |
| All-cause mortality | 38 |
| **Secondary outcome** | |
| **CVD** | 13 |
| Heart failure | 8 |
| Myocardial infarction | 4 |
| Arrhythmias | 1 |
| Stroke | 0 |
| **Non-CVD** | 25 |
| Infections | 9 |
| Cancer | 7 |
| Senility | 4 |
| Renal failure | 3 |
| Liver disease | 0 |
| Others | 2 |
| **HF hospitalization** | 24 |
| **Non-fatal MI** | 2 |
| **Non-fatal stroke** | 6 |

Data are presented as the number of patients (n). CVD, Cardiovascular death; FIB-4, HF, Heart failure; MI, myocardial infarction

**Table 4. Cox proportional hazard analysis of all-cause mortality.**

| All-cause mortality | Hazard ratio (95% CI) | P-value |
|---|---|---|
| **Model 1: Unadjusted** | | |
| Tertile 1 (FIB-4 index ≤ 2.17) | Reference | Reference |
| Tertile 2 (2.18 ≤ FIB-4 index ≤ 3.28) | 0.81 (0.25–2.57) | 0.73 |
| Tertile 3 (FIB-4 index ≥ 3.29) | 4.75 (2.05–11.0) | <0.001 |
| **Model 2: Adjusted for sex and ALBI score** | | |
| Tertile 1 (FIB-4 index ≤ 2.17) | Reference | Reference |
| Tertile 2 (2.18 ≤ FIB-4 index ≤ 3.28) | 0.68 (0.21–2.18) | 0.52 |
| Tertile 3 (FIB-4 index ≥ 3.29) | 3.67 (1.51–8.90) | 0.003 |
| **Model 3: Adjusted for sex, LBBB at baseline, paced-QRSd, and Vp rate at final check** | | |
| Tertile 1 (FIB-4 index ≤ 2.17) | Reference | Reference |
| Tertile 2 (2.18 ≤ FIB-4 index ≤ 3.28) | 0.84 (0.26–2.71) | 0.78 |
| Tertile 3 (FIB-4 index ≥ 3.29) | 4.79 (2.04–11.2) | <0.001 |
| **Model 4: Adjusted for sex, BNP, eGFR, and TR-PG** | | |
| Tertile 1 (FIB-4 index ≤ 2.17) | Reference | Reference |
| Tertile 2 (2.18 ≤ FIB-4 index ≤ 3.28) | 0.66 (0.19–2.18) | 0.49 |
| Tertile 3 (FIB-4 index ≥ 3.29) | 3.48 (1.38–8.78) | 0.007 |
| **Model 5: Adjusted for ALBI score, BNP, and Hb** | | |
| Tertile 1 (FIB-4 index ≤ 2.17) | Reference | Reference |
| Tertile 2 (2.18 ≤ FIB-4 index ≤ 3.28) | 0.50 (0.15–1.67) | 0.26 |
| Tertile 3 (FIB-4 index ≥ 3.29) | 2.10 (0.79–5.55) | 0.13 |

Data are presented as hazard ratios and 95% confidence intervals. ALBI, Albumin–bilirubin; BNP, Brain natriuretic protein; CI, Confidence interval; eGFR, estimated glomerular filtration rate; FIB-4, Fibrosis-4; LBBB, Left bundle branch block; QRSd, QRS duration; TR-PG, Tricuspid regurgitant pressure gradient; Vp, Ventricular pacing.

and BNP level (FIB-4 index: $P < 0.001$; Fig 1A, BNP: $P = 0.008$; Fig 2). However, due to suspected deviations from proportional hazards during this period (proportional hazards test: FIB-4 index, $P = 0.09$; BNP, $P = 0.035$), reassessment was performed with an extended observation period of up to 1000 days.

As a result, over approximately three years, only the FIB-4 index showed statistical significance (FIB-4 index, $P = 0.0014$; BNP, $P = 0.16$). In Cox hazard analysis, comparing Tertile 3 to Tertile 1 as the reference, the hazard ratio for the FIB-4 index was 3.68 ($P = 0.010$), while for BNP was 2.87 ($P = 0.070$).

Considering the significant impact of age on mortality, we computed the FIB-4 index without its inclusion (modified FIB-4 index = [AST] level / (platelets × square root [ALT] level)) and performed Cox analysis. The results revealed a hazard ratio of 3.64 (95% CI: 1.88–7.07, $P < 0.001$) for all-cause mortality (modified FIB-4 calculated after logarithmic transformation).

**Table 5. Cox proportional hazard analysis of all-cause mortality adjusted for ALBI score, BNP, and hemoglobin.**

| All-cause mortality | Hazard ratio (95% CI) | *P*-value |
|---|---|---|
| **Model 1: Adjusted for ALBI score** | | |
| **Tertile 1 (FIB-4 index ≤ 2.17)** | Reference | Reference |
| **Tertile 2 (2.18 ≤ FIB-4 index ≤ 3.28)** | 0.67 (0.21–2.13) | 0.49 |
| **Tertile 3 (FIB-4 index ≥ 3.29)** | 3.25 (1.33–7.91) | 0.009 |
| **ALBI score** | 2.87 (1.33–6.19) | 0.007 |
| **Model 2: Adjusted for BNP** | | |
| **Tertile 1 (FIB-4 index ≤ 2.17)** | Reference | Reference |
| **Tertile 2 (2.18 ≤ FIB-4 index ≤ 3.28)** | 0.63 (0.19–2.09) | 0.45 |
| **Tertile 3 (FIB-4 index ≥ 3.29)** | 3.35 (1.36–8.24) | 0.008 |
| **BNP** | 1.00 (1.00–1.00) | 0.020 |
| **Model 3: Adjusted for hemoglobin** | | |
| **Tertile 1 (FIB-4 index ≤ 2.17)** | Reference | Reference |
| **Tertile 2 (2.18 ≤ FIB-4 index ≤ 3.28)** | 0.67 (0.21–2.14) | 0.50 |
| **Tertile 3 (FIB-4 index ≥ 3.29)** | 3.18 (1.29–7.84) | 0.011 |
| **Hemoglobin** | 0.78 (0.63–0.96) | 0.020 |
| **Model 4: Adjusted for ALBI score and BNP** | | |
| **Tertile 1 (FIB-4 index ≤ 2.17)** | Reference | Reference |
| **Tertile 2 (2.18 ≤ FIB-4 index ≤ 3.28)** | 0.55 (0.16–1.81) | 0.32 |
| **Tertile 3 (FIB-4 index ≥ 3.29)** | 2.54 (1.00–6.45) | 0.048 |
| **ALBI score** | 3.27 (1.22–8.75) | 0.018 |
| **BNP** | 1.00 (0.99–1.00) | 0.094 |
| **Model 5: Adjusted for ALBI score and Hb** | | |
| **Tertile 1 (FIB-4 index ≤ 2.17)** | Reference | Reference |
| **Tertile 2 (2.18 ≤ FIB-4 index ≤ 3.28)** | 0.61 (0.19–1.95) | 0.40 |
| **Tertile 3 (FIB-4 index ≥ 3.29)** | 2.64 (1.04–6.71) | 0.040 |
| **ALBI score** | 2.28 (1.00–5.20) | 0.048 |
| **Hemoglobin** | 0.84 (0.67–1.05) | 0.14 |
| **Model 6: Adjusted for BNP and hemoglobin** | | |
| **Tertile 1 (FIB-4 index ≤ 2.17)** | Reference | Reference |
| **Tertile 2 (2.18 ≤ FIB-4 index ≤ 3.28)** | 0.56 (0.16–1.85) | 0.34 |
| **Tertile 3 (FIB-4 index ≥ 3.29)** | 2.55 (0.99–6.55) | 0.051 |
| **BNP** | 1.00 (0.99–1.00) | 0.074 |
| **Hemoglobin** | 0.81 (0.65–1.01) | 0.070 |
| **Model 7: Adjusted for ALBI score, BNP, and hemoglobin** | | |
| **Tertile 1 (FIB-4 index ≤ 2.17)** | Reference | Reference |
| **Tertile 2 (2.18 ≤ FIB-4 index ≤ 3.28)** | 0.50 (0.15–1.67) | 0.26 |
| **Tertile 3 (FIB-4 index ≥ 3.29)** | 2.10 (0.79–5.55) | 0.13 |
| **ALBI score** | 2.82 (1.03–7.74) | 0.042 |
| **BNP** | 1.00 (0.99–1.00) | 0.12 |
| **Hemoglobin** | 0.85 (0.68–1.07) | 0.18 |

Data are presented as hazard ratios and 95% confidence intervals. ALBI, Albumin–bilirubin; BNP, Brain natriuretic protein; CI, Confidence interval; FIB-4, Fibrosis-4.

The evaluation of the predictive accuracy of the FIB-4 index for all-cause mortality was performed using ROC curve analysis, resulting in an area under the curve of 0.72 (95% CI: 0.62–0.82). The cut-off value for the FIB-4 index was 3.75, with a specificity of 0.86, and a sensitivity

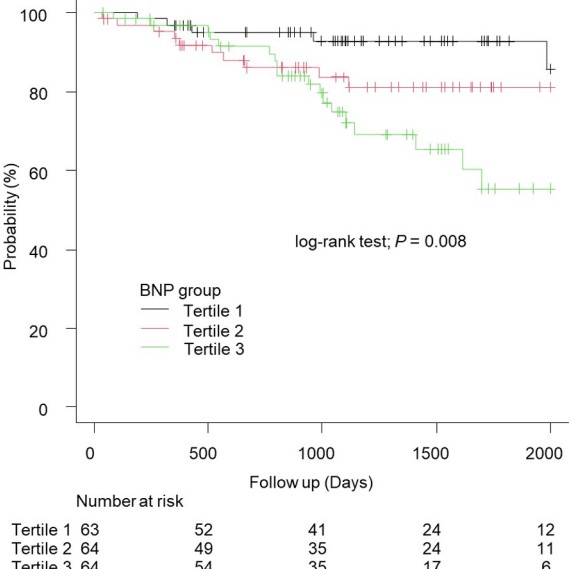

**Fig 2. Kaplan–Meier curve of all-cause mortality stratified by BNP tertiles.** BNP, brain natriuretic protein.

of 0.63 (Fig 3A). In addition, the FIB-4 index and its individual components were compared to assess their index values in predicting mortality, and the AUCs for ALT, AST, age, and platelet count were 0.45, 0.54, 0.61, and 0.64, respectively (each $P$ value compared to FIB-4; ALT $< 0.001$, AST $= 0.001$, age $= 0.073$, and platelet count $= 0.099$; Fig 3B).

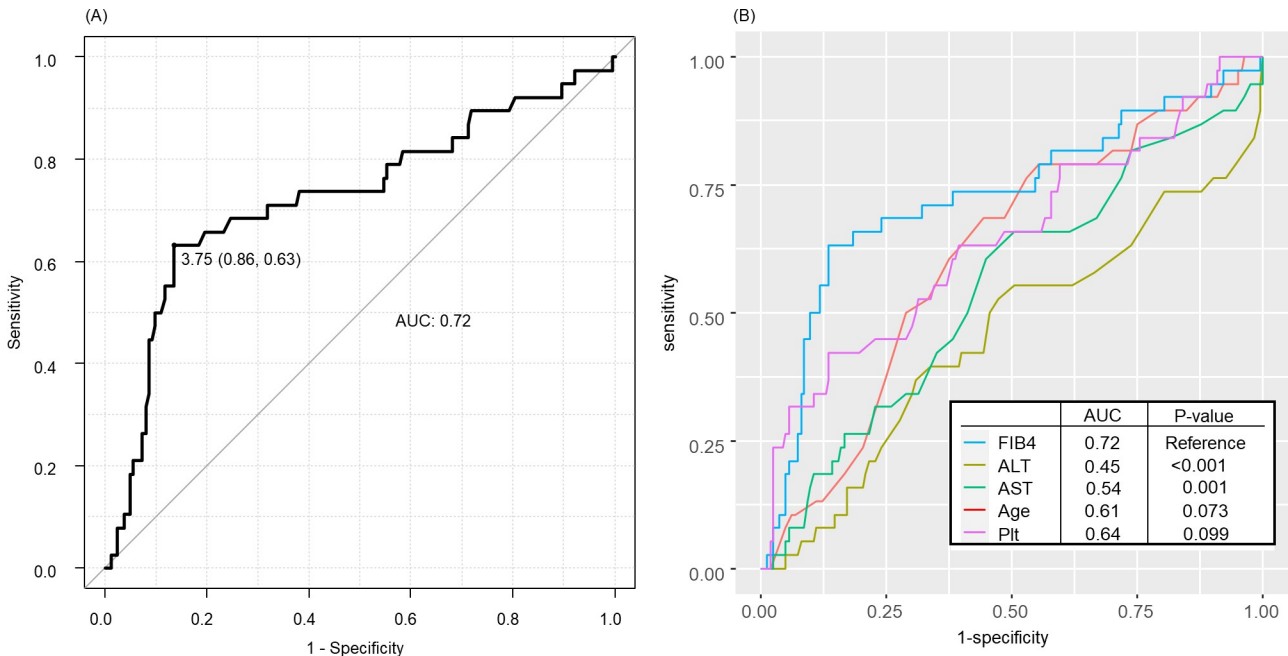

**Fig 3. ROC curve analysis of the association between the FIB-4 index and the risk of all-cause mortality.** (A) ROC curve for the FIB-4 index (B) Comparison of the FIB-4 index and its components as predictors of mortality. ALT, alanine aminotransferase; AST, aspartate aminotransferase; ROC, receiver operating characteristic; AUC, area under the curve; Plt, Platelets.

Next, using the cut-off value of the FIB-4 index calculated from the ROC curve, we classified the population into two groups: FIB-4 index <3.75 (Low FIB-4 group) and ≥3.75 (High FIB-4 group). Subsequently, we conducted log-rank tests using Kaplan-Meier analysis. The results from the log-rank test showed significant differences between the two groups for all-cause mortality, cardiovascular death, and non-cardiovascular death (all-cause mortality: $P < 0.001$, cardiovascular death: P = 0.0024, non-cardiovascular death: $P < 0.001$; Fig 4A–4C). In contrast, no significant difference was observed for MACE ($P = 0.11$; Fig 4D).

Finally, Cox analysis was conducted for all-cause mortality between the Low group and the High group (Table 6). The results differed from those of the tertile analysis; even after adjusting for ALBI score, BNP, and hemoglobin, the significance of the FIB-4 index as a predictor remained consistent (High FIB-4 group; HR = 4.85, 95% CI: 2.38–9.88, $P < 0.001$).

In the subgroup analysis of the association between all-cause mortality and the FIB-4 index, consistent results were observed (Fig 5).

## Discussion

This study showed that in patients with RVP implants, a high FIB-4 index was associated with an increased risk of all-cause mortality. This trend remained consistent even after adjusting for pacemaker-related factors. However, concerns have arisen about the value of the FIB-4 index when adjusted for ALBI score, BNP, and hemoglobin. Consequently, upon comparing the significance of the FIB-4 index and BNP as predictive factors, the superiority of the FIB-4 index in predicting overall mortality over approximately three years was evident. Furthermore, the predictive accuracy of the FIB-4 index for overall mortality demonstrated moderate precision, with a cut-off value of 3.75. These findings are novel contributions of this study.

A meta-analysis compared cardiac resynchronization therapy and His/left bundle area pacing [13]; however, there is still a lack of studies specifically focusing on RVP and all-cause mortality. Spath et al. reported a major composite endpoint, which included all-cause mortality, related to the lead position of RVP [14]; the all-cause mortality rate during the approximately 1000-day follow-up period was approximately 20% of the analyzed participants, which was similar to the death event rate in this study [14]. Subgroup analysis indicated that clinical outcomes improved when the QRS duration during pacing was narrow (< 130 ms) compared with when it was wide (≥ 130 ms) [14]. Therefore, we hypothesized that pacemaker-related factors may significantly affect mortality in patients with pacemakers and performed a multivariate analysis with pacing QRS duration and ventricular pacing rate as confounding factors.

We also evaluated the association between the FIB-4 index and cardiovascular death as a secondary outcome. As mentioned earlier, RVP may significantly affect left ventricular contraction and valvular pathology due to its non-physiological pacing pattern, increasing the risk of heart failure and atrial fibrillation [1–3]. Considering that the FIB-4 index may reflect the pathophysiology of circulatory failure, we hypothesized that there would be a significant association between the FIB-4 index and cardiovascular death. Although the Kaplan–Meier analysis showed no statistical significance, we believe that this might have been due to the extremely low number of cardiovascular death events during the 3-year follow-up period.

Notably, advancements in medical therapy for heart failure have reduced ejection fraction, transcatheter aortic valve replacement [15], MitraClip procedures for valvular disease [16], and interventional treatments for myocardial infarction. These factors, along with the widespread adoption of ablation therapy for atrial fibrillation, indicate that long-term follow-up is necessary for further analysis of cardiovascular death. Therefore, although no statistical differences in cardiovascular death were observed, the results of future studies are anticipated to shed light on the clinical significance of the FIB-4 index. On the other hand, heart failure

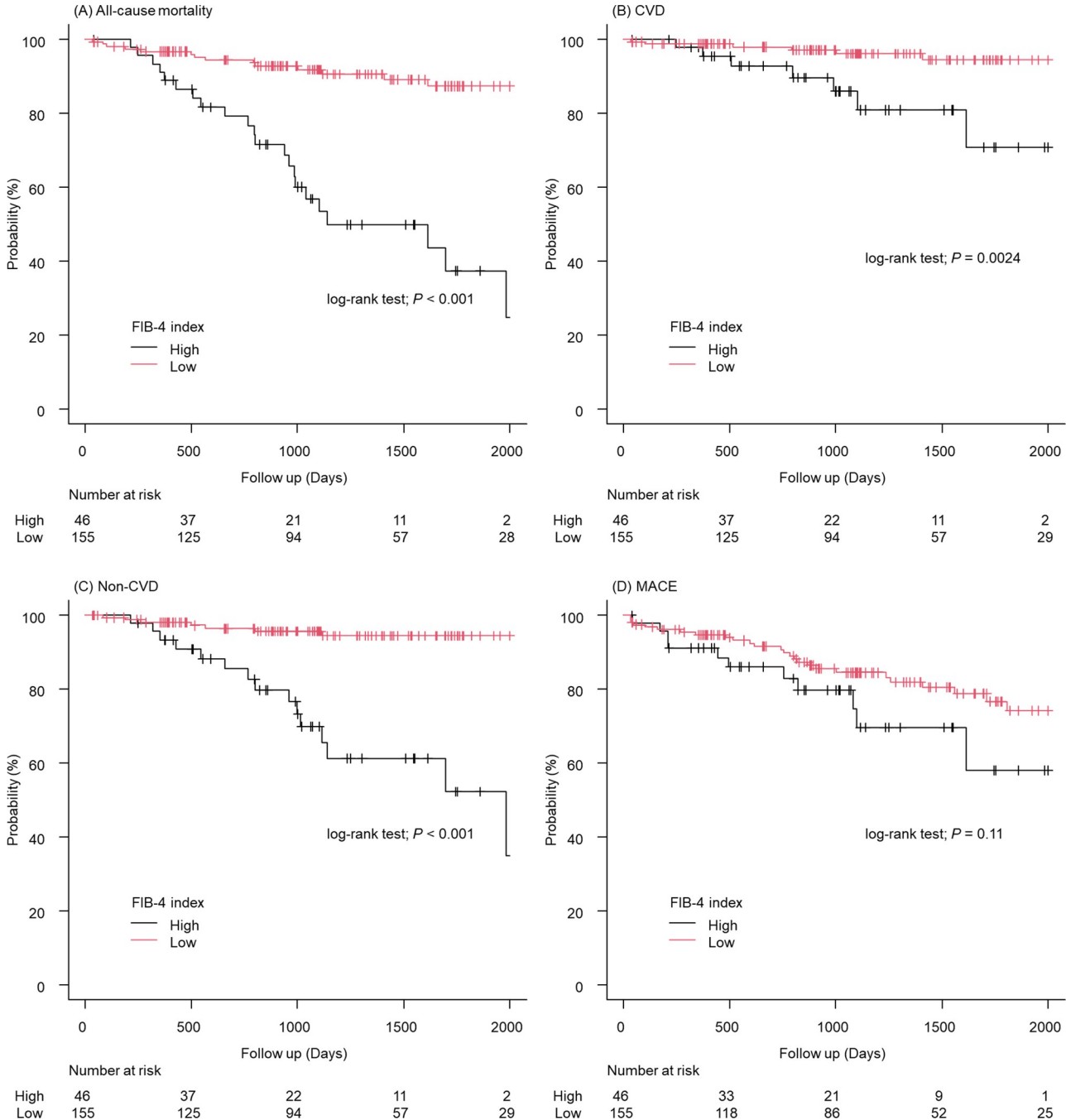

**Fig 4. All-cause mortality and secondary outcomes stratified by the cutoff value of the FIB-4 index.** (A) Kaplan–Meier curve of all-cause mortality. (B) Cardiovascular death. (C) Non-cardiovascular death. (D) MACE. CVD, Cardiovascular death; FIB-4, Fibrosis-4; MACE, Major adverse cardiovascular events.

hospitalization was more frequent in other major adverse cardiovascular events, while deaths related to infections and cancer were more prevalent in non-CVD. The relationship between the FIB-4 index and heart failure has been reported, specifically suggesting a potential overlap of risk factors or mechanisms between heart failure preserved ejection fraction and liver fibrosis [17]. This association is speculated to remain unaffected, even after pacemaker

**Table 6. Cox proportional hazard analysis of all-cause mortality between two groups.**

| All-cause mortality | Hazard ratio (95% CI) | P-value |
|---|---|---|
| **Model 1: Unadjusted** | | |
| FIB-4 index ≥ 3.75 | 7.46 (3.82–14.5) | <0.001 |
| **Model 2: Adjusted for ALBI score** | | |
| FIB-4 index ≥ 3.75 | 6.34 (3.23–12.5) | <0.001 |
| ALBI score | 3.78 (1.64–8.71) | 0.0017 |
| **Model 3: Adjusted for BNP** | | |
| FIB-4 index ≥ 3.75 | 5.83 (2.91–11.6) | <0.001 |
| BNP | 1.00 (1.00–1.00) | 0.031 |
| **Model 4: Adjusted for hemoglobin** | | |
| FIB-4 index ≥ 3.75 | 6.25 (3.16–12.3) | <0.001 |
| Hemoglobin | 0.75 (0.61–0.91) | 0.004 |
| **Model 5: Adjusted for ALBI score and BNP** | | |
| FIB-4 index ≥ 3.75 | 5.17 (2.56–10.4) | <0.001 |
| ALBI score | 3.42 (1.26–9.23) | 0.015 |
| BNP | 1.00 (0.99–1.00) | 0.14 |
| **Model 6: Adjusted for ALBI score and hemoglobin** | | |
| FIB-4 index ≥ 3.75 | 5.89 (2.97–11.6) | <0.001 |
| ALBI score | 2.76 (1.12–6.78) | 0.26 |
| Hemoglobin | 0.82 (0.66–1.02) | 0.078 |
| **Model 7: Adjusted for BNP and hemoglobin** | | |
| FIB-4 index ≥ 3.75 | 5.24 (2.59–10.6) | <0.001 |
| BNP | 1.00 (0.99–1.00) | 0.20 |
| Hemoglobin | 0.79 (0.64–0.97) | 0.030 |
| **Model 8: Adjusted for ALBI, BNP, and hemoglobin** | | |
| FIB-4 index ≥ 3.75 | 4.85 (2.38–9.88) | <0.001 |
| ALBI score | 2.74 (0.97–7.66) | 0.054 |
| BNP | 1.00 (0.99–1.00) | 0.25 |
| Hemoglobin | 0.84 (0.67–1.05) | 0.12 |

Data are presented as hazard ratios and 95% confidence intervals. ALBI, Albumin–bilirubin; BNP, Brain natriuretic protein; CI, Confidence interval; FIB-4, Fibrosis-4.

implantation. In non-CVD, deaths related to liver diseases were not observed, and none of the patients had indications of liver-related cancer at baseline. Nevertheless, in patients with a high FIB-4 index, there is a hypothetical possibility of an increased risk of non-CVD, suggesting the presence of underlying causes leading to liver injury and the systemic impact of inflammatory cytokines due to repetitive injuries [17, 18]. However, it is crucial to note that this study is not designed to elucidate these factors, and therefore, these observations remain speculative.

This study has some limitations. It was a retrospective single-center study; therefore, the influence of potential biases, such as an imbalance in the patient population, cannot be denied. Moreover, owing to the limited number of deaths, multivariate analysis with sufficient consideration of confounding factors could not be conducted. In addition, this study included patients with sinoatrial node dysfunction, in whom the degree of fusion pacing may be affected by the delay settings, which could potentially affect clinical outcomes. Furthermore, although the ventricular pacing rate is an important factor in clinical outcomes, this study only

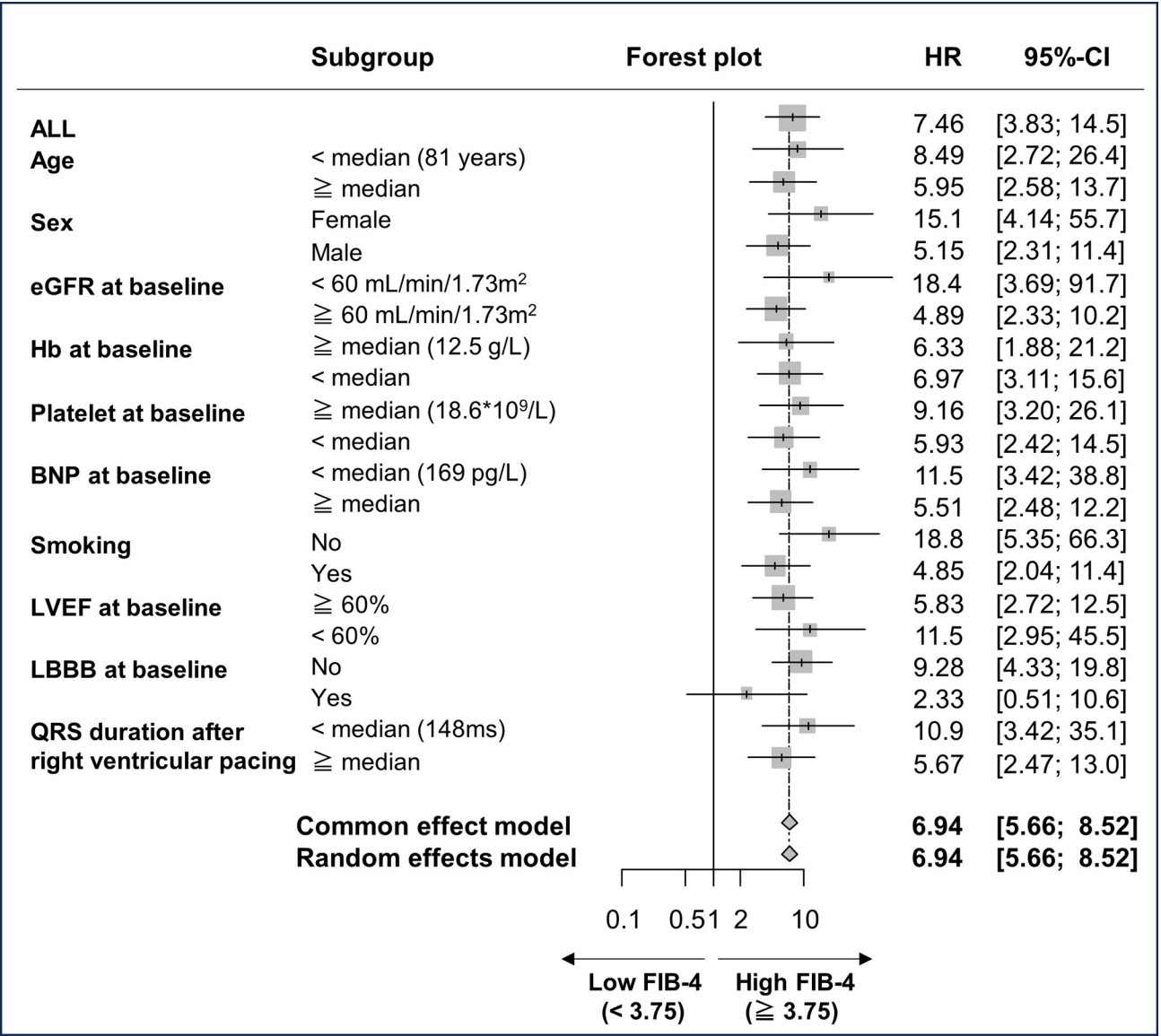

**Fig 5. Forest plots from the subgroup analysis assessing the FIB-4 index and incidence of all-cause mortality.** BNP, B-type natriuretic peptide; CI, Confidence interval; eGFR, estimated glomerular filtration rate; FIB-4, Fibrosis-4; Hb, Hemoglobin; HR, Hazard ratio; LBBB, Left bundle branch block; LVEF, Left ventricular ejection fraction.

considered the pacing rate at the final follow-up. In reality, there may be long-term variations in pacing rates for each patient, and it is necessary to investigate the relationship between temporal pacing rate changes and death events. Finally, as this study retrospectively evaluated patients who underwent pacemaker implantation, accurate information regarding the presence of liver diseases (such as cancer or viral hepatitis) and assessments (such as the Child-Pugh classification) could not be obtained. Consequently, we were unable to establish exclusion criteria based on these factors or to consider the impact of known liver diseases on the results of this study.

## Conclusion

In summary, the FIB-4 index may be a potential predictor of early all-cause mortality in patients with right ventricular pacemakers.

## Supporting information

**S1 Table. Anonymous dataset of 201 patients.**
(XLSX)

## Acknowledgments

We are grateful to Dr. Takahiro Imaizumi and Dr. Yuji Ito for their helpful discussions and comments on this manuscript.

## Author Contributions

**Conceptualization:** Naoya Inoue, Takehiro Hiramatsu, Toyoaki Murohara.

**Data curation:** Takashi Ogane, Takehiro Hiramatsu.

**Formal analysis:** Shuji Morikawa, Takehiro Hiramatsu.

**Investigation:** Takehiro Hiramatsu.

**Methodology:** Naoya Inoue.

**Project administration:** Naoya Inoue.

**Supervision:** Toyoaki Murohara.

**Validation:** Toyoaki Murohara.

**Visualization:** Naoya Inoue.

**Writing – original draft:** Naoya Inoue, Toyoaki Murohara.

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
