## [Decision Letter · Decision Letter 0]

18 Dec 2023

PONE-D-23-34285Clinical value of the fibrosis-4 index in predicting mortality in patients with right ventricular pacingPLOS ONE

Dear Dr. Inoue,

Thank you for submitting your manuscript to PLOS ONE. After careful consideration, we feel that it has merit but does not fully meet PLOS ONE’s publication criteria as it currently stands. Therefore, we invite you to submit a revised version of the manuscript that addresses the points raised during the review process.

We look forward to receiving your revised manuscript.

Kind regards,

Zhehao Dai

Academic Editor

PLOS ONE

Journal Requirements:

Reviewers' comments:

Reviewer's Responses to Questions

**Comments to the Author**

1. Is the manuscript technically sound, and do the data support the conclusions?

Reviewer #1: Partly

Reviewer #2: Partly

2. Has the statistical analysis been performed appropriately and rigorously? 

Reviewer #1: Yes

Reviewer #2: No

3. Have the authors made all data underlying the findings in their manuscript fully available?

Reviewer #1: No

Reviewer #2: Yes

4. Is the manuscript presented in an intelligible fashion and written in standard English?

Reviewer #1: Yes

Reviewer #2: Yes

5. Review Comments to the Author

Reviewer #1: The authors represented the association between FIB-4 index and all-cause mortality in patients implanted pacing device. The article was well-written, but I have several revisions as below.

Major revisions.

- The fibrosis-4 index is originally established to evaluate the liver fibrosis in patients with liver disease, such as HCV/HIV or NAFLD. On the other hands, recent reports indicated that the value of fibrosis-4 index may reflect hemodynamic condition in ADHF phase. Thus, this index at the time of pacemaker implantation may reflect both potential liver fibrosis degree and increased liver congestion due to bradycardia. Thus, the information of prevalence of liver disease such as viral hepatitis or NAFLD should add the baseline characteristics, and the influence of presence of liver disease should be evaluated by such as multivariate analysis or subgroup analysis.

- Fibosis-4 index includes age in its calculation. Especially, age is a very important factor for prognosis in any disease. Thus, the influence of age on prognostic impact of fibrosis-4 index should be carefully evaluated. So, the additional analysis, such as subgroup analysis divided by age or same analysis by the value of fibrosis-4 index calculated without age, should be considered to evaluate the impact of age for mortality. In addition, the prognostic value of each single factor included in calculation of fibrosis-4 index should be added.

Minor revision

- The details of CVD and non-CVD should be provided. Especially, it is important whether the significant association between high FIB-4 and non-CVD was derived by liver-related death.

Reviewer #2: Dr. Inoue et al. conducted an interdisciplinary retrospective study on the FIB-4 index as a predictor of mortality in patients with right ventricular pacing. This is an interesting perspective and the study contains a relatively large sample size. However, I am concerned with some aspects of the study design and the clinical implications are unclear. My comments are as follows.

Major:

1. The authors are proposing a cut-off FIB-4 index of 3.75 in their ROC curve. Therefore, they should present data in 2 groups (FIB-4 index >= 3.75 and < 3.75), not on the top and bottom tertiles. (Also, there is no significant difference between the first and second tertiles, perhaps suggesting that they can be combined.)

2. Exclusion criteria: why weren’t patients with liver diseases (in particular, cirrhosis) excluded? If not excluding such patients, parameters relating to liver status should be added to Table 1, such as presence/absence of liver disease and Child-Pugh or mALBI scores. It also seems that patients with ITP or other diseases that affect FIB-4 index that are unrelated to right ventricular function were not excluded. Patients with known cancer also have a short prognosis and should have been excluded.

3. In Figure 1, the median follow-up is 3 years, but most patients do not die within 3 years after pacemaker implantation. There is also a very large number of censored events. In fact, only 13 of 201 patients experienced CVD-related death in the study. What were the causes of death in the 25 patients who had non-CVD death? Where they related to the pacemaker or to heart function? Did they have liver disease? List causes of death in detail, as this is the primary outcome.

4. While age will no doubt affect prognosis, do the authors truly believe that laboratory results at pacemaker implantation, most of which were within the normal range, predict death several years later (mostly due to non-CVD events)? If so, what are the clinical implications of a high FIB-4 at the time of pacemaker implantation? These should also be addressed in the discussion section. Perhaps an analysis of the 38 deaths (compared to the other 163 cases) is warranted as a separate table.

5. Along the same lines, perhaps MACE should have bene considered as the primary outcome, or at least as a secondary outcome.

6. In Table 1, there is no significant difference in ALT between groups. Cox analysis shows that FIB-4 has independent predictive value even when the impact of age is excluded. The difference in AST at baseline is significant, but minimal and possibly not clinically relevant. Thus, it may appear that platelets are the key determining factor. Was FIB-4 a better predictor than platelets alone? Alternatively, perhaps age and platelets are adequate to predict mortality (not AST and ALT)? Please discuss which individual components of FIB-4 and their relevance for prognosis in this population.

7. Table 3: Instead of “primary outcome,” state the actual primary outcome (all-cause mortality) so make it easier to understand for the reader.

8. Table 3: why is the variable “sex” included in models 2 to 4? It is not significantly different between groups. Please clarify in your Methods section the criteria for inclusion in the multivariate Cox analysis. Because of the small number of events (deaths), included variables must be selected carefully, based on objective criteria.

9. Table 3: Also, should “age” be included in the multivariate Cox analysis? It is a part of the FIB-4 index.

10. Line 138: “No significant differences were observed in terms of baseline treatment, cardiac ultrasound examination results, or pacemaker parameters at the last follow-up.” I do not believe this is correct; there was a significant different in TRPG between groups.

11. Figure 2: Is the Forest plot comparing tertile 1 and tertile 3? This should be clearly mentioned. Also, at the bottom of the graph, the authors should show which direction favors tertile 1 and which side favors tertile 3.

12. In Figure 2, the overall hazard ratio (HR) for “ALL” is 1.43. However the unadjusted HR for Tertile 3 vs. Tertile 1 in Table 3 is 4.75. Please explain this discrepancy.

13. A more in-depth discussion is necessary, including discussions on the abovementioned issues.

Minor:

1. Introduction: “In a study targeting heart failure in patients with preserved left ventricular systolic function, those with a FIB-4 index ≥ 3.11 had a 2.202-fold (95% confidence interval: 1.110–4.368) higher risk of major adverse cardiovascular events (composite of cardiovascular death, heart failure-related rehospitalization, non-fatal myocardial infarction, and non-fatal stroke) [10].” Higher risk compared to what? Those with FIB-4 index < 3.11? Please state the comparative arm.

2. Line 95: “…deaths resulting from fatal arrhythmias, fatal myocardial infarction, heart failure, and stroke…” In this sentence, the word “fatal” (used twice) is not necessary in either instance because the authors are talking about “deaths” in the first place.

3. Line 135: “Significant differences were observed in various elements of the FIB-4 index,

4. including age, AST, platelet count, eGFR, hemoglobin, low-density lipoprotein cholesterol, BNP, and dysplipidemia” Only age, AST, and platelet count are elements of the index. Please rephrase.

6. PLOS authors have the option to publish the peer review history of their article (what does this mean?). If published, this will include your full peer review and any attached files.

Reviewer #1: **Yes: **Mitsutaka Nakashima

Reviewer #2: **Yes: **Takeshi Okamoto

---

## [Author Response · Author response to Decision Letter 0]

28 Dec 2023

December 26, 2023

Emily Chenette

Editor-in-Chief

PLOS ONE

Dear Editor: 

I wish to re-submit the manuscript titled “Clinical value of the fibrosis-4 index in predicting mortality in patients with right ventricular pacing.” The manuscript ID is PONE-D-23-34285.

We thank you and the reviewers for your thoughtful suggestions and insights. The manuscript has benefited from these insightful suggestions. I look forward to working with you and the reviewers to move this manuscript closer to publication in the PLOS ONE.

The manuscript has been rechecked and the necessary changes have been made in accordance with the reviewers’ suggestions. The revisions to the main text have been highlighted in yellow. The responses to all comments have been prepared and attached herewith. 

Thank you for your consideration. I look forward to hearing from you.

Sincerely,

Naoya Inoue

Department of Cardiology, Chutoen General Medical Center

1-1 Shobugaike, Kakegawa, Shizuoka, Japan

Phone No: 090-6581-8948

Fax No: 0537-28-8971

Email Address: inouenaoya251410@gmail.com

 

Editorial Office Comments:

Response:

Thank you for your comments. We have revised the manuscript based on the template provided. Additionally, we have requested proofreading to ensure compliance with the style requirements.  

Reviewer #1: The authors represented the association between FIB-4 index and all-cause mortality in patients implanted pacing device. The article was well-written, but I have several revisions as below.

Major revisions.

1- The fibrosis-4 index is originally established to evaluate the liver fibrosis in patients with liver disease, such as HCV/HIV or NAFLD. On the other hands, recent reports indicated that the value of fibrosis-4 index may reflect hemodynamic condition in ADHF phase. Thus, this index at the time of pacemaker implantation may reflect both potential liver fibrosis degree and increased liver congestion due to bradycardia. Thus, the information of prevalence of liver disease such as viral hepatitis or NAFLD should add the baseline characteristics, and the influence of presence of liver disease should be evaluated by such as multivariate analysis or subgroup analysis.

Response:

Thank you very much for your invaluable guidance. We appreciate your observations, and, as you pointed out, we agree that in this study, it was important to consider information about liver diseases, including setting the exclusion criteria accordingly, and to also assess the impact of viral hepatitis and NAFLD, which were factors that should have been considered for exclusion.

However, the retrospective nature of the study design and its focus on patients who underwent pacemaker implantation has led to some limitations. Consequently, there was limited information available regarding liver diseases in the study population. Additionally, there was a lack of sufficient blood test information to score the presence of ascites and the severity of liver diseases. Reliable data that are necessary for excluding cancer and other diseases could not be obtained. Acknowledging the significance of your comments, and in line with similar observations from Reviewer 2, we have obtained additional data specifically for the ALBI score and have included this information in the manuscript. Furthermore, we have added a note regarding these limitations in the "Limitations" section of the manuscript.

“Finally, as this study retrospectively evaluated patients who underwent pacemaker implantation, accurate information regarding the presence of liver diseases (such as cancer or viral hepatitis) and assessments (such as the Child-Pugh classification) could not be obtained. Consequently, we were unable to establish exclusion criteria based on these factors or consider the impact of known liver diseases on the results of this study.”

2- Fibosis-4 index includes age in its calculation. Especially, age is a very important factor for prognosis in any disease. Thus, the influence of age on prognostic impact of fibrosis-4 index should be carefully evaluated. So, the additional analysis, such as subgroup analysis divided by age or same analysis by the value of fibrosis-4 index calculated without age, should be considered to evaluate the impact of age for mortality. In addition, the prognostic value of each single factor included in calculation of fibrosis-4 index should be added.

Response:

As per your pertinent comments, we have performed an additional ROC analysis of each component. The results and a comparison of each component with the FIB-4 index as a reference were shown in Fig 2.

Furthermore, as you rightly pointed out, we agree that age has a significant impact on prognosis. Therefore, we calculated the Fib-4 index without considering age (modified FIB-4: AST/(Platelets × sqrt(ALT))) and conducted Cox analysis. The results revealed a hazard ratio of 3.64 (95%CI: 1.88–7.07, P<0.001) for overall mortality (modified FIB-4 calculated after logarithmic transformation). Recognizing the importance of this aspect, we have incorporated these findings into the main text (preceding Table 3).

“Considering the significant impact of age on mortality, we computed the FIB-4 index without its inclusion (modified FIB-4 index = [AST] level / (platelets × square root [ALT] level)) and performed Cox analysis. The results revealed a hazard ratio of 3.64 (95% CI: 1.88–7.07, P < 0.001) for all-cause mortality (modified FIB-4 calculated after logarithmic transformation).”

Minor revision

3- The details of CVD and non-CVD should be provided. Especially, it is important whether the significant association between high FIB-4 and non-CVD was derived by liver-related death.

Response:

We apologize for the lack of information. We have added the relevant text (S2 Table).

Reviewer #2: Dr. Inoue et al. conducted an interdisciplinary retrospective study on the FIB-4 index as a predictor of mortality in patients with right ventricular pacing. This is an interesting perspective and the study contains a relatively large sample size. However, I am concerned with some aspects of the study design and the clinical implications are unclear. My comments are as follows.

Major:

1. The authors are proposing a cut-off FIB-4 index of 3.75 in their ROC curve. Therefore, they should present data in 2 groups (FIB-4 index >= 3.75 and < 3.75), not on the top and bottom tertiles. (Also, there is no significant difference between the first and second tertiles, perhaps suggesting that they can be combined.)

Response:

Thank you for your important comment. As you pointed out, we acknowledge the significance of performing a data analysis using a cutoff value from our study population. Therefore, we have added supplemental data from a Kaplan-Meier analysis with log-rank tests, conducted by categorizing the cohort into two groups: FIB-4 index <3.75 (Low FIB-4 group) and ≥3.75 (High FIB-4 group).

2. Exclusion criteria: why weren’t patients with liver diseases (in particular, cirrhosis) excluded? If not excluding such patients, parameters relating to liver status should be added to Table 1, such as presence/absence of liver disease and Child-Pugh or mALBI scores. It also seems that patients with ITP or other diseases that affect FIB-4 index that are unrelated to right ventricular function were not excluded. Patients with known cancer also have a short prognosis and should have been excluded.

Response:

Thank you very much for your invaluable guidance. We appreciate and agree with your observations that, in this study, it was necessary to consider information on liver diseases, the establishment of exclusion criteria based on them, and the evaluation of the impact of ITP (immune thrombocytopenia) and known cancer patients, as you rightly pointed out.

However, the retrospective nature of the study design and its focus on patients who underwent pacemaker implantation has led to some limitations. Consequently, there was limited information available regarding liver diseases in the study population. Additionally, there was a lack of sufficient blood test information to score the presence of ascites and the severity of liver diseases. Reliable data that are necessary for excluding cancer and other diseases could not be obtained. 

We have revisited and supplemented the data, focusing specifically on the ALBI score (Table1, 3). Furthermore, we have described the aforementioned points in the section discussing the study limitations.

“Finally, as this study retrospectively evaluated patients who underwent pacemaker implantation, accurate information regarding the presence of liver diseases (such as cancer or viral hepatitis) and assessments (such as the Child-Pugh classification) could not be obtained. Consequently, we were unable to establish exclusion criteria based on these factors or consider the impact of known liver diseases on the results of this study.”

3. In Figure 1, the median follow-up is 3 years, but most patients do not die within 3 years after pacemaker implantation. There is also a very large number of censored events. In fact, only 13 of 201 patients experienced CVD-related death in the study. What were the causes of death in the 25 patients who had non-CVD death? Where they related to the pacemaker or to heart function? Did they have liver disease? List causes of death in detail, as this is the primary outcome.

Response:

We apologize for the lack of information. We have added the relevant text (S2 Table).

4. While age will no doubt affect prognosis, do the authors truly believe that laboratory results at pacemaker implantation, most of which were within the normal range, predict death several years later (mostly due to non-CVD events)? If so, what are the clinical implications of a high FIB-4 at the time of pacemaker implantation? These should also be addressed in the discussion section. Perhaps an analysis of the 38 deaths (compared to the other 163 cases) is warranted as a separate table.

Response:

Thank you very much for your valuable question. Firstly, as you pointed out, we did not initially consider the possibility of predicting future mortality based on the mostly normal blood test results (or FIB-4 index) at the time of pacemaker implantation until the initiation of this study. However, the study primarily referenced, conducted by Saito et al. (Impact of the Fibrosis-4 Index on Risk Stratification of Cardiovascular Events and Mortality in Patients with Atrial Fibrillation: Findings from a Japanese Multicenter Registry. J Clin Med. 2020 Feb 21;9(2):584.), suggested the involvement of liver diseases in the onset and promotion of atrial fibrillation. In this study, patients with atrial fibrillation and a FIB-4 index of 2.51 or higher showed a significant increase in all-cause mortality compared to those with a FIB-4 index below 2.51 during a median follow-up period of 39.2 months. However, the detailed causes of death were not specified in relation to the primary outcomes, which included stroke, major bleeding, and cardiovascular events.

Given the context of such previous research, patients predominantly characterized by bradycardiac arrhythmias requiring pacemaker implantation are likely to have a high incidence of concomitant conditions such as congestive hepatopathy and right heart failure. While other liver diseases may also contribute, we assumed that cases with elevated FIB-4 index are often influenced by congestion. Partially supporting this assumption is the finding that the group with a higher FIB-4 index tended to have higher BNP and baseline TR-PG in the comparison of patient characteristics. However, there are several limitations that prevent us from establishing the significance of this study. These include the inability to assess the occurrence of new arrhythmias post pacemaker implantation and the fact that the majority of pacemaker patients did not undergo blood tests during follow-up. Therefore, we were unable to investigate changes in the FIB-4 index after implantation. Ideally, if the presence or absence of liver disease at the time of pacemaker implantation could be accurately evaluated, we would be able to explore the relationship between the baseline FIB-4 index level and its cause. Furthermore, we would be able to accurately assess its impact on life expectancy, as you rightly pointed out.

5. Along the same lines, perhaps MACE should have bene considered as the primary outcome, or at least as a secondary outcome.

Response:

As you pointed out, we have added MACE (cardiovascular death, hospitalization for heart failure, non-fatal myocardial infarction, or non-fatal stroke) as a secondary outcome and appended the analysis results in the revised main text (Table 2, Fig1, Fig 3).

6. In Table 1, there is no significant difference in ALT between groups. Cox analysis shows that FIB-4 has independent predictive value even when the impact of age is excluded. The difference in AST at baseline is significant, but minimal and possibly not clinically relevant. Thus, it may appear that platelets are the key determining factor. Was FIB-4 a better predictor than platelets alone? Alternatively, perhaps age and platelets are adequate to predict mortality (not AST and ALT)? Please discuss which individual components of FIB-4 and their relevance for prognosis in this population.

Response:

As per your important comments, we have performed an additional ROC analysis of each component. The results and a comparison of each component with the FIB-4 index as a reference are shown in Fig 2.

7. Table 3: Instead of “primary outcome,” state the actual primary outcome (all-cause mortality) so make it easier to understand for the reader.

Response:

We have made the correction regarding Table 3 as you pointed out.

8. Table 3: why is the variable “sex” included in models 2 to 4? It is not significantly different between groups. Please clarify in your Methods section the criteria for inclusion in the multivariate Cox analysis. Because of the small number of events (deaths), included variables must be selected carefully, based on objective criteria.

9. Table 3: Also, should “age” be included in the multivariate Cox analysis? It is a part of the FIB-4 index.

Response:

Thank you for your highly valuable feedback. Firstly, regarding the inclusion of age as an adjusting factor, we acknowledge, as you rightly pointed out, that it was inappropriate as it is already a component of the FIB-4 index. We sincerely apologize for this oversight. In response, we have excluded age as an adjusting factor and conducted the Cox analysis once again.

Concerning the inclusion of sex as an adjusting factor, we would like to explain the rationale behind this decision. For variable inclusion in this study, we opted for the forced entry method, as indicated in the Methods section. The variables chosen for forced entry were based on their clinical significance to the objectives of this study, drawing from prior research and universal concepts. Notably, when considering mortality as an outcome, it is worth mentioning that the average lifespan differs between men and women in the Japanese population, as indicated on the Ministry of Health, Labour and Welfare's website (https://www.mhlw.go.jp/content/10904750/000872952.pdf).

Furthermore, in a prior study focused on Japanese patients with atrial fibrillation (AF) utilizing the FIB-4 index (Saito Y, et al. Impact of the Fibrosis-4 Index on Risk Stratification of Cardiovascular Events and Mortality in Patients with Atrial Fibrillation: Findings from a Japanese Multicenter Registry. J Clin Med. 2020 Feb 21;9(2):584.), sex was reported as a significant factor associated with mortality. Hence, considering the importance of sex in the context of mortality in studies involving Japanese individuals with AF, we decided to include it as an adjusting factor in our study.

However, as you rightly highlighted, we agree that the careful selection of adjusting factors is crucial, particularly given the limited number of death events in our study. We appreciate your insight on this matter.

10. Line 138: “No significant differences were observed in terms of baseline treatment, cardiac ultrasound examination results, or pacemaker parameters at the last follow-up.” I do not believe this is correct; there was a significant different in TRPG between groups.

Response:

We apologize for the inaccuracies in the statement. We have made the necessary corrections based on your observations.

“No significant differences were observed in terms of treatment at baseline or pacemaker parameters at the last follow-up. However, there was a significant difference in the TR-PG on the cardiac ultrasound examination and the proportion of left bundle branch block (LBBB) on the baseline ECG”

11. Figure 2: Is the Forest plot comparing tertile 1 and tertile 3? This should be clearly mentioned. Also, at the bottom of the graph, the authors should show which direction favors tertile 1 and which side favors tertile 3.

12. In Figure 2, the overall hazard ratio (HR) for “ALL” is 1.43. However the unadjusted HR for Tertile 3 vs. Tertile 1 in Table 3 is 4.75. Please explain this discrepancy.

13. A more in-depth discussion is necessary, including discussions on the abovementioned issues.

Response:

Thank you for your incredibly important feedback. Your comments have brought to our attention an error in our subgroup analysis, and for this oversight, we sincerely apologize.

In the subgroup analysis, we mistakenly treated the FIB-4 index as a continuous variable, leading to discrepancies with the results presented in Table 3 (where the FIB-4 index was categorized into tertiles). Therefore, following your guidance (which was also mentioned at Q1) we have opted for a comparative analysis using the cutoff value obtained from the ROC curve (categorizing FIB-4 index as High FIB-4 for values ≥3.75 and Low FIB-4 for values <3.75). We have represented these comparisons in the figure for the subgroup analysis, indicating the favorable directions accordingly. Additionally, we have made the necessary amendments and clarifications in the Results section and Fig 4. We sincerely apologize for any confusion caused.

Minor:

1. Introduction: “In a study targeting heart failure in patients with preserved left ventricular systolic function, those with a FIB-4 index ≥ 3.11 had a 2.202-fold (95% confidence interval: 1.110–4.368) higher risk of major adverse cardiovascular events (composite of cardiovascular death, heart failure-related rehospitalization, non-fatal myocardial infarction, and non-fatal stroke) [10].” Higher risk compared to what? Those with FIB-4 index < 3.11? Please state the comparative arm.

Response:

Thank you for your suggestion. Upon reviewing the reference materials once again, we have made additional amendments to the text as follows.

“patients with an FIB-4 index ≥ 3.11 had a 2.202-fold (95% confidence interval: 1.110–4.368) higher risk of major adverse cardiovascular events (composite of cardiovascular death, heart failure-related rehospitalization, non-fatal myocardial infarction, and non-fatal stroke) compared to patients with an index <3.11 after adjustment for male sex, serum creatinine level, and hemoglobin concentration.”

2. Line 95: “…deaths resulting from fatal arrhythmias, fatal myocardial infarction, heart failure, and stroke…” In this sentence, the word “fatal” (used twice) is not necessary in either instance because the authors are talking about “deaths” in the first place.

Response:

According to your suggestion, we have removed the term 'fatal'.

3. Line 135: “Significant differences were observed in various elements of the FIB-4 index, including age, AST, platelet count, eGFR, hemoglobin, low-density lipoprotein cholesterol, BNP, and dysplipidemia” Only age, AST, and platelet count are elements of the index. Please rephrase.

Response:

We apologize for the inaccurate expression. As you pointed out, we have revised the sentence as follows.

“In addition to the elements of the FIB-4 index, such as age, AST, and platelet count, significant differences were observed in the eGFR, hemoglobin, low-density lipoprotein cholesterol, and BNP levels, and dyslipidemia.”

---

## [Decision Letter · Decision Letter 1]

3 Jan 2024

PONE-D-23-34285R1Clinical value of the fibrosis-4 index in predicting mortality in patients with right ventricular pacingPLOS ONE

Dear Dr. Inoue,

Thank you for submitting your revised manuscript to PLOS ONE. After careful consideration, we feel that it has merit but does not fully meet PLOS ONE’s publication criteria as it currently stands. Therefore, we invite you to submit a further revised version of the manuscript that addresses the points raised during the review process.

We look forward to receiving your revised manuscript.

Kind regards,

Zhehao Dai

Academic Editor

PLOS ONE

Journal Requirements:

Editor Comments:

The authors were able to revise the manscript according to most comments of the reviewers. Please address the newly raised concerns by reviewer#2, after which we would like to consider acceptance.

Reviewers' comments:

Reviewer's Responses to Questions

**Comments to the Author**

1. If the authors have adequately addressed your comments raised in a previous round of review and you feel that this manuscript is now acceptable for publication, you may indicate that here to bypass the “Comments to the Author” section, enter your conflict of interest statement in the “Confidential to Editor” section, and submit your "Accept" recommendation.

Reviewer #1: All comments have been addressed

Reviewer #2: (No Response)

2. Is the manuscript technically sound, and do the data support the conclusions?

Reviewer #1: Yes

Reviewer #2: Partly

3. Has the statistical analysis been performed appropriately and rigorously? 

Reviewer #1: Yes

Reviewer #2: No

4. Have the authors made all data underlying the findings in their manuscript fully available?

Reviewer #1: Yes

Reviewer #2: Yes

5. Is the manuscript presented in an intelligible fashion and written in standard English?

Reviewer #1: Yes

Reviewer #2: Yes

6. Review Comments to the Author

Reviewer #1: The authors revised appropriately. I agree acceptance of this report.

Reviewer #2: Dr. Inoue et al. have significantly improved their manuscript based on reviewers’ suggestions. My remaining comments are as follows.

Major:

1. I appreciate the limitation added, but I am concerned that the study does not sufficiently consider liver disease when using a liver index to predict outcomes in the field of cardiology. I understand that information on liver function is limited, but I believe chart reviews can reveal medical histories at the time of pacemaker implantation, at least with respect to hepatitis B and hepatitis C (from blood tests), and maybe history of liver disease and alcohol use.

2. Table 1: There are 3 groups, so please clarify what the P-values refer to. Is this a 3-group comparison or a comparison between Tertiles 1 and 3? If the former, there is no description of multiple comparisons in the statistical analysis section.

3. Thank you for adding S2 Table. I believe this is an important table and may be included in the main text, unless there is an upper limit on the number of tables/figures. Core aspects can be included in the Results and Discussion sections of the main text. In particular, 16 of the 38 deaths were due to cancer or infection. Given death within 3 years, some of the 9 patients who died of cancer probably had cancer at the time of pacemaker implantation. This is why baseline information is very important. The authors have very detailed information on CV risk factors in Table 1, so I believe medical charts at the time of pacemaker implantation will reveal any known history of cancer at that time. Cancer is also a CV risk factor.

4. It is reassuring to see that FIB-4 was a significant predictor independent of ALBI. However, the grouping of factors in the multivariable Cox analysis still appears arbitrary to me. I would like to see a multivariable Cox analysis including the following 4 factors:

FIB-4, ALBI (or albumin), hemoglobin, and BNP. These were the factors with the lowest P values in Table 1. I would like to know if FIB-4 remains an independent predictor in this analysis.

5. As stated previously, I believe the Discussion should be expanded to discuss various aspects of the study, including possible relationship with non-cardiovascular deaths. The authors have added new analyses as per the reviewers’ suggestions, but have not discussed their new analyses in the Discussion section.

Minor:

1. “Ultimately, 201 patients were included in the study.” is stated under Materials and Methods, but this is a result. It is also mentioned at the beginning of the results section, so it can be deleted from Materials and Methods.

2. EZR software: provide the version number.

3. Table 1: “ECG after RVP implant” should be “ECG after RVP implantation”

4. Table 1: “Atrium” should be “Atrial” to be consistent with “Ventricular” which comes up later on.

5. S2 Table: Provide a footnote with abbreviations. Also, it would be easier to interpret if the causes of death were ordered by frequency.

7. PLOS authors have the option to publish the peer review history of their article (what does this mean?). If published, this will include your full peer review and any attached files.

Reviewer #1: **Yes: **Mitsutaka Nakashima

Reviewer #2: **Yes: **Takeshi Okamoto

---

## [Author Response · Author response to Decision Letter 1]

5 Jan 2024

January 6, 2024

Emily Chenette

Editor-in-Chief

PLOS ONE

Dear Editor: 

I wish to re-submit the manuscript titled “Clinical value of the fibrosis-4 index in predicting mortality in patients with right ventricular pacing.” The manuscript ID is PONE-D-23-34285.

We thank you and the reviewers for your thoughtful suggestions and insights. The manuscript has benefited from these insightful suggestions. I look forward to working with you and the reviewers to move this manuscript closer to publication in the PLOS ONE.

The manuscript has been rechecked and the necessary changes have been made in accordance with the reviewers’ suggestions. The revisions to the text have been highlighted in yellow. The responses to all comments have been prepared and attached herewith. 

Thank you for your consideration. I look forward to hearing from you.

Sincerely,

Naoya Inoue

Department of Cardiology, Chutoen General Medical Center

1-1 Shobugaike, Kakegawa, Shizuoka, Japan

Phone No: 090-6581-8948

Fax No: 0537-28-8971

Email Address: inouenaoya251410@gmail.com

 

Editorial Office Comments:

Response:

We have checked the reference list and found no retracted papers among the cited articles.

 

Reviewer #1: The authors revised appropriately. I agree acceptance of this report.

Response:

Thank you very much for taking the time to review our paper amid your busy schedule. We sincerely appreciate your valuable feedback and comments. 

 

Reviewer #2: Dr. Inoue et al. have significantly improved their manuscript based on reviewers’ suggestions. My remaining comments are as follows.

Major:

1. I appreciate the limitation added, but I am concerned that the study does not sufficiently consider liver disease when using a liver index to predict outcomes in the field of cardiology. I understand that information on liver function is limited, but I believe chart reviews can reveal medical histories at the time of pacemaker implantation, at least with respect to hepatitis B and hepatitis C (from blood tests), and maybe history of liver disease and alcohol use.

Response:

Thank you very much for your invaluable comments. We also apologize for the lack of information regarding liver diseases. As you pointed out, we have added additional details regarding the results of hepatitis virus testing at the time of pacemaker insertion. Regarding alcohol consumption, we decided to omit this information owing to limited reliability in the medical records, including the frequency, amount, and, in some cases, the presence of alcohol consumption. We sincerely apologize for the inconvenience.

"The number of patients with a positive HCV Ab was significantly different among the three groups; however, post hoc analysis using the Bonferroni method revealed no significant differences between each pair of groups (Tertile 1 vs 2; P = 1, Tertile 1 vs 3; P = 0.091, Tertile 2 vs 3; P = 0.23).” 

2. Table 1: There are 3 groups, so please clarify what the P-values refer to. Is this a 3-group comparison or a comparison between Tertiles 1 and 3? If the former, there is no description of multiple comparisons in the statistical analysis section.

Response:

Thank you for your comment. We apologize for the lack of clarity in our explanation. We have added the following sentences to the Statistical Analysis section:

"In the comparison of patient characteristics among three groups, we conducted the Kruskal-Wallis test and reported the P-value for overall comparison. To adjust for the significance level in multiple comparisons, we performed post-hoc analyses using Bonferroni correction."

3. Thank you for adding S2 Table. I believe this is an important table and may be included in the main text, unless there is an upper limit on the number of tables/figures. Core aspects can be included in the Results and Discussion sections of the main text. In particular, 16 of the 38 deaths were due to cancer or infection. Given death within 3 years, some of the 9 patients who died of cancer probably had cancer at the time of pacemaker implantation. This is why baseline information is very important. The authors have very detailed information on CV risk factors in Table 1, so I believe medical charts at the time of pacemaker implantation will reveal any known history of cancer at that time. Cancer is also a CV risk factor.

Response:

Thank you for your valuable suggestion. We agree with you that the table in S2 Table is important, as you have indicated. Therefore, we have added it as a new Table 3 in the text instead of in the supplemental data. In addition, we have added “Cancer” to the CV risk factors in Table 1.

“Regarding cancer, a cancer related to the liver was not observed as one of the CV risk factors. While a significant overall difference was noted for cancer, Bonferroni analysis showed no significant differences between Tertile 1 vs 2; P = 1, Tertile 1 vs 3; P = 0.62, Tertile 2 vs 3; P = 0.076."

4. It is reassuring to see that FIB-4 was a significant predictor independent of ALBI. However, the grouping of factors in the multivariable Cox analysis still appears arbitrary to me. I would like to see a multivariable Cox analysis including the following 4 factors:

FIB-4, ALBI (or albumin), hemoglobin, and BNP. These were the factors with the lowest P values in Table 1. I would like to know if FIB-4 remains an independent predictor in this analysis.

Response:

Thank you for your valuable comments. Following your suggestion, we have conducted an additional analysis using the factor with the lowest p-value in Table 1. In this analysis, no statistically significant difference was observed in the hazard ratio between Tertile3 and Tertile1. However, we consider this result to be highly important; therefore, we have added the following text to Table 4:

“Furthermore, the hazard ratio was 2.10 (95% CI: 0.79–5.55, P = 0.13) when adjusted for ALBI score, BNP, and Hb, which were significantly different in the comparison of patient characteristics (Table 4).”

5. As stated previously, I believe the Discussion should be expanded to discuss various aspects of the study, including possible relationship with non-cardiovascular deaths. The authors have added new analyses as per the reviewers’ suggestions, but have not discussed their new analyses in the Discussion section.

Response:

Thank you for your comments. We apologize for the insufficient discussion in our part. We have added insights regarding this new analysis in the Discussion section.

“On the other hand, heart failure hospitalization was more frequent in other major adverse cardiovascular events, while deaths related to infections and cancer were more prevalent in Non-CVD. The relationship between the FIB-4 index and heart failure has been reported, specifically suggesting a potential overlap of risk factors or mechanisms between heart failure preserved ejection fraction and liver fibrosis [17]. This association is speculated to remain unaffected even after pacemaker implantation. In Non-CVD, deaths related to liver diseases were not observed, and none of the patients had indications of liver-related cancer at baseline. Nevertheless, in patients with high FIB-4 index, there is a hypothetical possibility of an increased risk of Non-CVD, suggesting the presence of underlying causes leading to liver injury and the systemic impact of inflammatory cytokines due to repetitive injuries [17,18]. However, it is crucial to note that this study is not designed to elucidate these factors, and therefore, these observations remain speculative.”

Minor:

1. “Ultimately, 201 patients were included in the study.” is stated under Materials and Methods, but this is a result. It is also mentioned at the beginning of the results section, so it can be deleted from Materials and Methods.

Response:

Thank you for your valuable opinion and apologize for this inaccurate expression. As pointed out, we have removed this text.

2. EZR software: provide the version number.

Response:

Thank you for pointing this out to us. We have added it.

3. Table 1: “ECG after RVP implant” should be “ECG after RVP implantation”

Response:

Thank you for your comments. We have corrected the word that you pointed out.

4. Table 1: “Atrium” should be “Atrial” to be consistent with “Ventricular” which comes up later on.

Response:

Thank you for your comments. We have corrected the word that you pointed out.

5. S2 Table: Provide a footnote with abbreviations. Also, it would be easier to interpret if the causes of death were ordered by frequency.

Response:

Thank you for your valuable advice. As mentioned earlier, we inserted the S2 table in Table 3 into the main text and arranged the causes of death in order of frequency. We have also added annotations for these abbreviations.

---

## [Decision Letter · Decision Letter 2]

15 Jan 2024

PONE-D-23-34285R2Clinical value of the fibrosis-4 index in predicting mortality in patients with right ventricular pacingPLOS ONE

Dear Dr. Inoue,

Thank you for submitting your manuscript to PLOS ONE. After careful consideration, we feel that it has merit but does not fully meet PLOS ONE’s publication criteria as it currently stands. Therefore, we invite you to submit a revised version of the manuscript that addresses the points raised during the review process.

We look forward to receiving your revised manuscript.

Kind regards,

Zhehao Dai

Academic Editor

PLOS ONE

Journal Requirements:

Additional Editor Comments:

I appreciate the effort and time that the authors spent on revisions of the paper. However, I agree with Reviewer#2 that the authors should be more cautious on the interpretatio when the ajusted results show no associations between FIB-4 and the endpoint after adjustment of ALBI, BNP, and hemoglobin, which were different among groups at baseline.

Would you please revise the interpretation of the results? For example, "However, the results of our study revealed that the FIB-4 index was independently associated with all-cause mortality" was a misinterpretation in this case. Please also discuss about the possible explaination why the association was neutralized by those adjustment.

Reviewers' comments:

Reviewer's Responses to Questions

**Comments to the Author**

1. If the authors have adequately addressed your comments raised in a previous round of review and you feel that this manuscript is now acceptable for publication, you may indicate that here to bypass the “Comments to the Author” section, enter your conflict of interest statement in the “Confidential to Editor” section, and submit your "Accept" recommendation.

Reviewer #2: (No Response)

2. Is the manuscript technically sound, and do the data support the conclusions?

Reviewer #2: Partly

3. Has the statistical analysis been performed appropriately and rigorously? 

Reviewer #2: No

4. Have the authors made all data underlying the findings in their manuscript fully available?

Reviewer #2: Yes

5. Is the manuscript presented in an intelligible fashion and written in standard English?

Reviewer #2: Yes

6. Review Comments to the Author

Reviewer #2: The authors have again significantly revised their manuscript. While I appreciate their hard work, the revisions have led to significant concerns on my part.

1. The authors found that FIB-4 was NOT an independent predictor of shorter OS when controlling for ALBI, BNP, and hemoglobin. The authors do not show which of these 3 factors were independent predictor of shorter OS, but this at least means that a model with albumin, bilirubin, BNP, and hemoglobin is a better predictor than that with AST, ALT, age, and platelets (the FIB-4 index). This is not discussed at all, but this result should not be taken lightly, as it may significantly undermine the value and conclusions of this whole study.

2. There are many instances where the authors emphasize findings that support their hypothesis and downplay findings that do not support their hypothesis. I am concerned that the manuscript may mislead the reader to believe that the FIB-4 index is more valuable than it really is in the context of patients undergoing right ventricular pacing.

3. As a result of the added analysis, the clinical implications of this study and the FIB-4 index in the studied cohort have also become unclear.

7. PLOS authors have the option to publish the peer review history of their article (what does this mean?). If published, this will include your full peer review and any attached files.

Reviewer #2: **Yes: **Takeshi Okamoto

---

## [Author Response · Author response to Decision Letter 2]

22 Jan 2024

January 21, 2024

Emily Chenette

Editor-in-Chief

PLOS ONE

Dear Editor: 

I wish to re-submit the manuscript titled “Clinical value of the fibrosis-4 index in predicting mortality in patients with right ventricular pacing.” The manuscript ID is PONE-D-23-34285.

We thank you and the reviewers for your thoughtful suggestions and insights. The manuscript has benefited from these insightful suggestions. I look forward to working with you and the reviewers to move this manuscript closer to publication in the PLOS ONE.

The manuscript has been rechecked and the necessary changes have been made in accordance with the reviewers’ suggestions. The revisions to the text have been highlighted in yellow. The responses to all comments have been prepared and attached herewith. 

Thank you for your consideration. I look forward to hearing from you.

Sincerely,

Naoya Inoue

Department of Cardiology, Chutoen General Medical Center

1-1 Shobugaike, Kakegawa, Shizuoka, Japan

Phone No: 090-6581-8948

Fax No: 0537-28-8971

Email Address: inouenaoya251410@gmail.com

 

Editorial Office Comments:

I appreciate the effort and time that the authors spent on revisions of the paper. However, I agree with Reviewer#2 that the authors should be more cautious on the interpretatio when the ajusted results show no associations between FIB-4 and the endpoint after adjustment of ALBI, BNP, and hemoglobin, which were different among groups at baseline.

Would you please revise the interpretation of the results? For example, "However, the results of our study revealed that the FIB-4 index was independently associated with all-cause mortality" was a misinterpretation in this case. Please also discuss about the possible explaination why the association was neutralized by those adjustment.

Response:

Thank you for your important comments. We agree with the significance of the observations made by Reviewer #2. Subsequently, we have attempted to address the identified issues and prepared a comprehensive response to Reviewer #2, approaching it from various angles and to the best of our ability. Thank you for your comment.

 

Reviewer #2: The authors have again significantly revised their manuscript. While I appreciate their hard work, the revisions have led to significant concerns on my part.

Major:

1. 1. The authors found that FIB-4 was NOT an independent predictor of shorter OS when controlling for ALBI, BNP, and hemoglobin. The authors do not show which of these 3 factors were independent predictor of shorter OS, but this at least means that a model with albumin, bilirubin, BNP, and hemoglobin is a better predictor than that with AST, ALT, age, and platelets (the FIB-4 index). This is not discussed at all, but this result should not be taken lightly, as it may significantly undermine the value and conclusions of this whole study.

2. There are many instances where the authors emphasize findings that support their hypothesis and downplay findings that do not support their hypothesis. I am concerned that the manuscript may mislead the reader to believe that the FIB-4 index is more valuable than it really is in the context of patients undergoing right ventricular pacing.

3. As a result of the added analysis, the clinical implications of this study and the FIB-4 index in the studied cohort have also become unclear.

Response:

Thank you very much for your exceptionally important comments. First, we sincerely appreciate the accuracy of your observations and apologize for the part of our conclusion that was phrased in a way that could lead to misunderstanding. Accordingly, we have revised and removed the relevant sections.

As an initial consideration in response to the above points, we began by detailing the results of Cox analyses for various combinations, including ALBI score, BNP, Hb, and others.

All-cause mortality Hazard ratio

(95% CI) P-value

Unadjusted

Tertile 1 (FIB-4 index ≤ 2.17)

Tertile 2 (2.18 ≤ FIB-4 index ≤ 3.28)

Tertile 3 (FIB-4 index ≥ 3.29) 

Reference

0.81 (0.25-2.57)

4.75 (2.05-11.0) 

Reference

0.73

<0.001

Model 1: Adjusted for ALBI score

Tertile 1 (FIB-4 index ≤ 2.17)

Tertile 2 (2.18 ≤ FIB-4 index ≤ 3.28)

Tertile 3 (FIB-4 index ≥ 3.29)

ALBI score 

Reference

0.67 (0.21-2.13)

3.25 (1.33-7.91)

2.87 (1.33-6.19) 

Reference

0.49

0.009

0.007

Model 2: Adjusted for BNP

 Tertile 1 (FIB-4 index≤ 2.17)

Tertile 2 (2.18 ≤ FIB-4 index ≤ 3.28)

Tertile 3 (FIB-4 index ≥ 3.29)

BNP 

Reference

0.63 (0.19-2.09)

3.35 (1.36-8.24)

1.00 (1.00-1.00) 

Reference

0.45

0.008

0.020

Model 3: Adjusted for Hb

Tertile 1 (FIB-4 index≤2.17)

Tertile 2 (2.18 ≤ FIB-4 index ≤ 3.28)

Tertile 3 (FIB-4 index ≥ 3.29)

 Hb 

Reference

0.67 (0.21-2.14)

3.18 (1.29-7.84)

0.78 (0.63-0.96) 

Reference

0.50

0.011

0.020

Model 4: Adjusted for ALBI score and BNP 

 Tertile 1 (FIB-4 index≤ 2.17)

Tertile 2 (2.18 ≤ FIB-4 index ≤ 3.28)

Tertile 3 (FIB-4 index ≥ 3.29)

 ALBI score

 BNP 

Reference

0.55 (0.16-1.81)

2.54 (1.00-6.45)

3.27 (1.22-8.75)

1.00 (0.99-1.00) 

Reference

0.32

0.048

0.018

0.094

Model 5: Adjusted for ALBI score and Hb 

 Tertile 1 (FIB-4 index≤ 2.17)

Tertile 2 (2.18 ≤ FIB-4 index ≤ 3.28)

Tertile 3 (FIB-4 index ≥ 3.29)

 ALBI score

 Hb 

Reference

0.61 (0.19-1.95)

2.64 (1.04-6.71)

2.28 (1.00-5.20)

0.84 (0.67-1.05) 

Reference

0.40

0.040

0.048

0.14

Model 6: Adjusted for BNP and Hb 

 Tertile 1 (FIB-4 index≤ 2.17)

Tertile 2 (2.18 ≤ FIB-4 index ≤ 3.28)

Tertile 3 (FIB-4 index ≥ 3.29)

 BNP

 Hb 

Reference

0.56 (0.16-1.85)

2.55 (0.99-6.55)

1.00 (0.99-1.00)

0.81 (0.65-1.01) 

Reference

0.34

0.051

0.074

0.070

Model 7: Adjusted for ALBI, BNP, and Hb 

 Tertile 1 (FIB-4 index≤ 2.17)

Tertile 2 (2.18 ≤ FIB-4 index ≤ 3.28)

Tertile 3 (FIB-4 index ≥ 3.29)

ALBI score

BNP

Hb 

Reference

0.50 (0.15-1.67)

2.10(0.79-5.55)

2.82 (1.03-7.74)

1.00 (0.99-1.00)

0.85 (0.68-1.07) 

Reference

0.26

0.13

0.042

0.12

0.18

Based on the above results, we also agree with your observation that there were influences from ALBI score, BNP, and Hb. Subsequently, in response to the points raised in Question 1, we conducted ROC analysis to compare the AUC and assess whether the models for ALBI score, BNP, and Hb are superior predictive factors for the FIB-4 index.

PropensityScore.GLM.1: ALBI+BNP+Hb model

As a result, the AUC and 95% confidence intervals for each were as follows: FIB-4 index: 0.727 (0.62-0.82) and ALBI+BNP+Hb model: 0.718 (0.61-0.81). The P-value for the AUC comparison was 0.86, indicating no significant superiority of the ALBI+BNP+Hb model as a predictive factor. However, we also observed that the FIB-4 index did not demonstrate superiority as a predictive factor, and we agree with your assessment.

Subsequently, in accordance with your opinion, we re-evaluated the clinical value of the FIB-4 index in the group undergoing right ventricular pacing. Upon considering the Cox model, when controlled for ALBI and Hb (Model 5), significant differences in FIB-4 remained. However, with the introduction of BNP in Model 6, the significance of the FIB-4 index was no longer observed.

Therefore, to investigate the differences when using the BNP and FIB-4 index as predictive factors, we conducted a comparative assessment using Kaplan-Meier analysis. We will present the results of the Kaplan-Meier analysis, stratifying BNP into tertiles, similar to the FIB-4 index.

The results of the log-rank test were P<0.001 for the FIB-4 index and P=0.008 for BNP, both of which indicated significant differences. However, focusing on the proportional hazards assumption from this graph, it appeared that the hazard changed over time for the proportional hazards test (FIB-4 index, P=0.09; BNP, P=0.035). Therefore, considering concerns about proportional hazards during this period, we conducted a reassessment specifically for a duration of 1000 days (approximately three years).

As a result, the P-values were 0.0014 for the FIB-4 index and 0.16 for BNP, indicating that over approximately 3 years, only the FIB-4 index showed statistical significance. In the Cox proportional hazards analysis, comparing Tetile3 to Tetile1 as the reference, the hazard ratio for the FIB-4 index was HR=3.68, P=0.010, and for BNP, HR=2.87, P=0.070. The proportional hazards assumption was also confirmed, and neither the FIB-4 index nor BNP showed rejection of proportional hazards (P=0.33).

Based on a comparison with BNP, the FIB-4 index was considered to have clinical value as a predictor of mortality, potentially being able to predict earlier mortality events.

In your first comment (Reviewer #2), you suggested the following:

" The authors are proposing a cut-off FIB-4 index of 3.75 in their ROC curve. Therefore, they should present data in 2 groups (FIB-4 index >= 3.75 and < 3.75), not on the top and bottom tertiles. (Also, there is no significant difference between the first and second tertiles, perhaps suggesting that they can be combined.)"

Therefore, we conducted a similar Cox analysis between the two groups using this cut-off value.

All-cause mortality Hazard ratio

(95% CI) P-value

Unadjusted

FIB-4 index ≥ 3.75 

7.46 (3.82-14.5) 

<0.001

Model 1:Adjusted for ALBI

FIB-4 index ≥ 3.75

ALBI score 

6.34 (3.23-12.5)

3.78 (1.64-8.71) 

<0.001

0.0017

Model 2: Adjusted for BNP

FIB-4 index ≥ 3.75

BNP 

5.83 (2.91-11.6)

1.00 (1.00-1.00) 

<0.001

0.031

Model 3: Adjusted for Hb

FIB-4 index ≥ 3.75

 Hb 

6.25 (3.16-12.3)

0.75 (0.61-0.91) 

<0.001

0.004

Model 4: Adjusted for ALBI score and BNP 

 FIB-4 index ≥ 3.75

 ALBI score

 BNP 

5.17 (2.56-10.4)

3.42 (1.26-9.23)

1.00 (0.99-1.00) 

<0.001

0.015

0.14

Model 5: Adjusted for ALBI score and Hb 

 FIB-4 index ≥ 3.75

 ALBI score

 Hb 

5.89 (2.97-11.6)

2.76 (1.12-6.78)

0.82 (0.66-1.02) 

<0.001

0.26

0.078

Model 6: Adjusted for BNP and Hb 

 FIB-4 index ≥ 3.75

 BNP

 Hb 

5.24 (2.59-10.6)

1.00 (0.99-1.00)

0.79 (0.64-0.97) 

<0.001

0.20

0.030

Model 7: Adjusted for ALBI, BNP, and Hb 

 FIB-4 index ≥ 3.75

ALBI score

BNP

Hb 

4.85 (2.38-9.88)

2.74 (0.97-7.66)

1.00 (0.99-1.00)

0.84 (0.67-1.05) 

<0.001

0.054

0.25

0.12

In this analysis, the predictive value of the FIB-4 index remained consistent even after adjusting for ALBI score, BNP, and Hb. Based on these findings, we have revised our conclusions as follows: 

' In patients with pacemakers, the FIB-4 index may have value as a predictor of earlier all-cause mortality, with a cutoff value of 3.75.' 

We attribute the clinical significance to this study and the FIB-4 index.

Your invaluable insights prompted us to reconsider the clinical significance of this research, leading to the revision of our conclusions. We sincerely appreciate your invaluable input and have incorporated it into our responses.

---

## [Editor Report · Decision Letter 3]

24 Jan 2024

Clinical value of the fibrosis-4 index in predicting mortality in patients with right ventricular pacing

PONE-D-23-34285R3

Dear Dr. Inoue,

We’re pleased to inform you that your manuscript has been judged scientifically suitable for publication and will be formally accepted for publication once it meets all outstanding technical requirements.

Kind regards,

Zhehao Dai

Academic Editor

PLOS ONE

Additional Editor Comments (optional):

Thank you for your huge efforts to improve the manuscript.

---

## [Editor Report · Acceptance letter]

26 Jan 2024

PONE-D-23-34285R3 

PLOS ONE

Dear Dr. Inoue, 

I'm pleased to inform you that your manuscript has been deemed suitable for publication in PLOS ONE. Congratulations! Your manuscript is now being handed over to our production team.

Kind regards, 

on behalf of

Dr. Zhehao Dai 

Academic Editor

PLOS ONE